# E-contact facilitated by conversational agents reduces interethnic prejudice and anxiety in Afghanistan
Sofia Sahab [1] ✉, Jawad Haqbeen [1], Rafik Hadfi[1], Takayuki Ito [1], Richard Eke Imade [2], Susumu Ohnuma [3] & Takuya Hasegawa[4]

Intergroup contact occurring through indirect means such as the internet has the potential to improve intergroup relationships and may be especially beneficial in high conflict situations. Here we conducted a three-timepoint online experiment to ascertain whether the use of a conversational agent in E-contact platforms could mitigate interethnic prejudices and hostility among Afghanistan's historically segregated and persistently conflictual ethnic groups. 128 Afghans of Pashtun, Tajik, and Hazara backgrounds were assigned to one of four E-contact conditions (control with no conversational agent and three experimental groups that varied in the conversational agent settings). Participants in the experimental conditions contributed more ideas and longer opinions and showed a greater reduction in outgroup prejudice and anxiety than those in the control group. These findings demonstrate that E-contact facilitated by a conversational agent can improve intergroup attitudes even in contexts characterized by a long history of intergroup segregation and conflict.

Intergroup interactions, which are necessary for the promotion of social stability and harmony in society, unfortunately also have the potential for conflicts, which may arise from differences in interests, values, phenotypes, and culture[1–3]. The intergroup contact hypothesis is a prominent theoretical framework for reducing intergroup bias, tensions, and hostilities[4–6]. Allport (1954)[7] hypothesized that increased contacts between different or antagonistic groups could assuage intergroup anxiety and apprehension, leading to a more favorable reconsideration of initial stereotypes[8,9]. Although Allport[7] focused on direct in-person contact as a mechanism for transforming intergroup prejudice and hostility, scholars have increasingly acknowledged the role of various indirect contact channels in generating positive intergroup relations[3,10–14].

Intergroup contact theory hypothesizes that contacts between otherwise antagonistic groups could positively transform previously held mutual bias, anxiety and hostility, thereby culminating in improved intergroup relations under certain optimal conditions, such as equal status, cooperation, shared superordinate goal and institutional or normative support[15,16]. Some scholars have argued that voluntary participation in the contact situation and the potential to nurture friendship beyond instantaneous interaction or contact are no less important conditions[9,17]. Pettigrew and Tropp (2006)[15] found that whether or not the optimal conditions suggested

by Allport are fulfilled, the effects of contacts on intergroup relations remains positive, except that the positive effects are more profound if these conditions accompany contacts.

However, despite the widespread appeal that the contact hypothesis has received in academic and policy circles, some scholars have cautioned against the credulous association of intergroup contact with positive outcomes[18,19]. For example, Mousa's (2020)[20,21] study showed that although intergroup contacts improved relationship measures between ingroup and outgroup members engaged in the intervention, the transferability of these transformed intergroup relationship measures beyond the intervention's direct targets remains doubtful and cumbersome. Similarly, proponents of the contact hypothesis have not paid adequate attention to the possibility that contacts may actually exacerbate rather than reduce prejudice, anxiety and avoidance, particularly in contexts characterized by ethno-racial segregation, systemic inequality and endemic conflicts[16,18,20]. Indeed, Barlow et al. (2012)[22] and Graf et al. (2014)[23] found that although positive contact experiences seemed to be more frequent than negative contact experiences, the predictive power of the latter on prejudice tends to be more salient. Scholars have attributed this greater tendency of negative contacts to escalate intergroup bias, distrust and hostility to the greater publicity received by negative contact experiences, particularly in segregated and

[1]Department of Social Informatics, Kyoto University, Kyoto, Japan. [2]Department of International Cooperation Studies, Nagoya University, Nagoya, Japan. [3]Department of Behavioral Science / Center for Experimental Research in Social Sciences, Hokkaido University, Sapporo, Japan. [4]Department of Computer Science, Nagoya Institute of Technology, Nagoya, Japan. ✉e-mail: sahab.sofia@i.kyoto-u.ac.jp

persistently conflictual intergroup context like the one we studied[24,25]. Consequently, the intergroup contact hypothesis has been criticized for the discrepancy between its conceptual and experimental representation of intergroup contacts and the nitty-gritty of contacts that different individuals and groups experience in their daily lives, thereby offering limited account of the historical, ideological and institutional processes that may facilitate or inhibit the attainment of optimal contact prerequisites[18,19].

Furthermore, the landscape of intergroup interactions has been transformed by computer-mediated technologies that bridge gaps among individuals and groups separated by sociocultural, institutional, or spatial factors[12,26]. This form of interaction offers multifaceted advantages, encompassing the establishment of secure environments, reduction of anxiety, surmounting geographical barriers, ensuring cost efficiency, promotion of equal status and intimate interaction, as well as the cultivation of cooperative efforts[8]. Thus the scholarship on the intergroup contact hypothesis now encompasses various indirect channels of contact[10,11,13,14].

One form of indirect contact that has received research and policy attention is electronic contact (E-contact)[1,5,12,27,28]. Here, individuals from different groups interact via online, virtual, and/or internet platforms that enable, for example, video conferencing or chatting. The use of the internet may bridge interactional gaps between various antagonistic ethno-racial groups in view of the barriers that may militate against the possibility of direct contact in places where rigid segregation norms and rules, protracted conflicts, spatial impediments and so on remain endemic[3,12,27–29].

Unlike the evidence on direct face-to-face contacts, the evidence regarding the optimal conditions for E-contact remains sparse, thus suggesting the need for more studies to improve our understanding of how technology-mediated contacts can affect intergroup processes and reduce prejudices and hostility[5,28–30]. Likewise, there is a dearth of research exploring the effects of intergroup contact in contexts such as Afghanistan, which is understandable given the volatile intergroup relations and the difficulties it may impose on researchers, relevant authorities, and the participants themselves[31,32].

Our aim is to apply the contact hypothesis to a non-Western and conflict-affected context, which remains comparatively under-researched in the literature on intergroup relations[3,20,33,34]. We also seek to address a major issue with the use of E-contact platforms--deficient supervision[5,33,35,36]. Condra and Linardi's (2019)[33] study in Afghanistan showed that unstructured and unsupervised intergroup contacts may escalate rather than assuage intergroup anxiety and stereotypes, suggesting that researchers should pay greater attention to the conditions surrounding intergroup contacts. Accordingly, we highlight here a particular condition, the use of a conversational agent (CA) to facilitate E-contact, which may yield more fruitful outcomes than E-contact occurring without a CA. CAs are artificial intelligence programs that engage in human-like conversations, using natural language[37]. Having a CA facilitate online discussions could be a cost-efficient and relatively unbiased strategy for improving E-contact interactions by mitigating the risks of unstructured and unsupervised E-contact interactions by guiding and moderating the conversation to achieve desired goals. We use a CA that does not represent any of the existing ethnic groups in Afghanistan and that can act as a sort of neutral authority figure to support the goal of harmonious multi-ethnic relationships. Although Allport's[7] original concept of authority support focused on institutional and normative backing, we considered CAs a modern application of this idea to the extent that it structures the discussions and functions to ensure a level playing field that enables participants to participate on equal terms and with mutual respect for the opinions of other.

Afghanistan is a diverse country with a population of around 40 million people as of 2021[38], comprising 14 recognized ethnic groups according to the country's 2004 constitution. The six major ethnic groups are Pashtun, Tajik, Hazara, Uzbek, Aimaq, and Turkman[39]. Ethnic groupings in Afghanistan are defined by a range of characteristics, including descent, language, religious sect, and location[32,39]. Broadly speaking, ethnic groups in Afghanistan are divided into tribal and non-tribal groups. Controversy persists over the actual population of each ethnic group because their representatives frequently inflate statistics since no comprehensive census figures have been published by successive administrations[39].

Pashtuns have dominated the political and administrative apparatuses of Afghanistan since 1747 (with the exception of 9 months rule by the Tajiks in 1929, the civil wars of 1992–1996, and the years of striving for democracy between 2001 and 2021), when Ahmad Shah Durrani, a Pashtun ethnic, unified the Pashtun tribes and founded modern Afghanistan[40]. This "false unification" was achieved through the repression and subjugation of other ethnic groups, especially in the 1880s when Abdur Rahman Khan, who was the Emir of Afghanistan between 1880 and 1901 targeted the Hazaras for decimation, expulsion, and displacement[41]. This strategy fostered hatred between different groups and instigated deep ethnic and religious polarization that persist till date[41].

During the 1992-1996 civil wars, ethnicity played a leading role in determining the legitimacy of political regimes[32], as almost all ethnic factions resorted to extra-judicial killings, torture, and sexual violence targeted at civilian members of rival ethnic groups[42]. The Taliban's forceful seizure of power in 1996 reinforced the dominance of Pashtuns[32] and clearly changed the conflict into a persistent power struggle with non-Pashtun ethnic groups[43]. Throughout the short-lived democratic period from 2001–2021, ethnic tensions endured, and political elites and ethno-political factions manipulated ethnic politics and heightened ethnic competition[44]. The Hazara community experienced targeted attacks, highlighting the delicate state of inter-ethnic relations[45]. Subsequently, with the Taliban assuming control of the Afghan government in August 2021, these tensions witnessed a significant escalation due to the Taliban's explicit demonstration of Pashtun ethno-nationalism[44]. This is apparent in their predominantly Pashtun composition and active promotion of the Pashto over Dari as illustrated by the extensive substitution of bilingual signage in government and public institutions with Pashto[44,46]. The rise in extrajudicial killings of individuals from other ethnic groups and the forced displacement of Hazaras from their homes across various provinces of Afghanistan became distressingly widespread, exacerbating social divisions[45].

These ongoing intergroup tensions in Afghanistan set the stage for our study. Given the increased risk of direct face-to-face and unsupervised intergroup interactions exacerbating ethnic bias in such a conflict-affected society like Afghanistan[33], we chose to utilize E-contact facilitated by a Conversational Agent (CA) that remains impartial and unaffiliated with any ethnic groupings in Afghanistan.

In the current study we examine the effects of using a CA as an E-contact facilitator to foster the reduction of outgroup anxiety and prejudice in Afghanistan, a non-Western, post-conflict context that has received less attention in existing literature[24].

The study spans over two discussion sessions: a synchronous and an asynchronous session. The synchronous session extended over two hours, while the asynchronous session spanned three days. This approach allowed us to assess the potential benefits derived from utilizing both synchronous and asynchronous communication in E-contact.

In the experiment, an equal number of Pashtun and non-Pashtun (Tajik/Hazara) participants were randomly assigned to form four-person discussion groups across two sessions. During the synchronous session, participants were required to collaboratively prepare a proposal for the frozen assets of Afghanistan to incorporate one of Allport's (1954)[7] optimal contact conditions-collaboration and teamwork. Subsequently, in the asynchronous session, participants were required to collaboratively prepare a draft policy proposal for upgrading informal settlements. We also explored the role of social engagement in our study by assessing dimensions such as the number of ideas generated and the length of opinions. This allowed us to examine how the CA's facilitation impacted participants' engagement.

We tested the following hypothesis:

H1: Intergroup/ethnic discussions in online platforms, which are facilitated by CAs are more likely to reduce mutual prejudice than unfacilitated online discussions.

As part of our research design, we also explored if differences in the settings programed for the CA influenced intergroup outcomes.

## Methods

### Participants

An a priori power analysis, using G*Power v3.1[47] was conducted to determine the sample size for a $2 \times 3$ ANOVA repeated measures, within-between interaction. With a small effect size (partial $\eta 2 = 0.02$, effect size $f = 0.14$), significance level of 0.05, and power of 0.95, the sample size was determined as 128 participants. We targeted the three major ethnic groups of Afghanistan (Pashtuns, Tajiks and Hazaras) with the goal of forming discussion groups of equal-sized Pashtuns and non-Pashtuns (i.e., Tajiks/Hazaras) participants.

We recruited the online participants through a respondent recruiting agency based in Afghanistan. The call for participation was announced by the agency in their online job portal. The registration questionnaires included a consent form, demographic questions, and questions to measure English proficiency, as the discussion required English proficiency.

Following the request for participation, 3021 participants registered between March 1 and 14, 2022. Among these, 1501 participants were removed, due to incomplete items (948), repeated registrations (77), not residing in Afghanistan (50), belonging to ethnic groups other than the three targeted ethnic groups (193) and lack of English proficiency (233) (Fig. 1). To align with the equal status principle of Allport's theory, we selected only those aged 23–37 and those having Bachelors or Masters' education. These age and educational groups comprised the largest percentage of other groups in the sample. With this criterion set, 1226 participants remained which were divided into six strata by ethnicity and sex: Pashtun males (556), Pashtun females (36), Tajik males (376), Tajik females (66), Hazara males (148) and Hazara females (44). Finally, 48 Pashtun males, 16 Pashtun females, 24 Tajik Males, 8 Tajik females, 24 Hazara males, 8 Hazara females were randomly selected from the strata and randomly assigned to the control and treatment groups (refer to Table 1 for demographics). These participants were then organized into discussion groups, each consisting of four members: two Pashtun and two non-Pashtun. As there were fewer female participants (only 146 out of 1226), we selected a minimum of one discussion group for each of the four conditions (No CA facilitation, normal CA, CA prompting for issues and CA prompting for ideas). The discussion groups were designed to be single-sex (see Fig. 1). Participants were compensated with $30/AFN 3000 for participating in the discussions (two hours synchronous and three days asynchronous in D-agree) and surveys (at T1, T2, and T3, using SurveyMonkey).

A series of $t$-tests and one-way ANOVA tests were conducted to examine whether there were significant differences in the pre-intervention measures between the treatment and control groups, indicating successful randomization. The results showed no statistically significant evidence for differences in intergroup prejudice scores ($t_{Welch}$ (70) = 1.842, $p = 0.070$, Cohen's $d = 0.327$, 95% CI [−0.076, 0.728], $BF_{10} = 0.665$), intergroup anxiety scores ($t_{Welch}$ (87) = 1.637, $p = 0.105$, Cohen's $d = 0.266$, 95% CI [−0.136, 0.667], $BF_{10} = 0.455$), English proficiency scores ($t(126) = -1.753$, $p = 0.082$, Cohen's $d = -0.358$, 95% CI [−0.760, 0.045], $BF_{10} = 0.830$), or general knowledge scores ($t(126) = 0.823$, $p = 0.412$, Cohen's $d = 0.168$, 95% CI [−0.233, 0.568], $BF_{10} = 0.290$) between the two groups. However, the Bayes Factors for these analyses ranged between 0.290 and 0.830, providing mostly inconclusive evidence regarding the presence or absence of differences.

To assess the randomization of subjects across the three CA setting conditions (i.e., normal CA facilitation, CA prompting for issues and CA prompting for ideas), we conducted a series of one-way ANOVA tests. The results revealed no evidence for statistically significant differences in intergroup prejudice ($F_{Welch}$ (2, 61) = 0.770, $p = 0.467$, $\eta^2 = 0.019$, 95% CI [0.000, 0.089], $BF_{10} = 0.195$), intergroup anxiety ($F$(2, 93) = 0.239, $p = 0.788$, $\eta^2 = 0.005$, 95% CI [0.000, 0.048], $BF_{10} = 0.115$), English proficiency ($F_{Welch}$(2, 58) = 1.393, $p = 0.256$, $\eta^2 = 0.042$, 95% CI [0.000, 0.131], $BF_{10} = 0.473$) and general knowledge ($F$(2, 93) = 0.021, $p = 0.979$, $\eta^2 = 0.000$, 95% CI [0.000, 0.000], $BF_{10} = 0.096$). The Bayes Factors (<0.3) further support evidence in favor of the null hypothesis. However, for English

proficiency, the Bayes Factor of 0.473 provides inconclusive evidence regarding the presence or absence of differences.

### Study design and data analysis

The study employed a randomized controlled trial (RCT) design to investigate the causal effects of CA facilitation on intergroup prejudice and intergroup anxiety. The participants were randomly assigned to two setting conditions: CA facilitation versus no CA facilitation. Three- measurements of intergroup prejudice and intergroup anxiety were taken per participant: before the intervention (Time 1), after 2 h of synchronous intergroup discussion (Time 2), and after 3 days of asynchronous intergroup discussion (Time 3).

Repeated measures ANOVA with between-subject factor (condition) and within-subject factor (time) was chosen as the appropriate statistical test to analyze the differences on the dependent variables (intergroup prejudice and intergroup anxiety). A significance level of 0.05 and two-tailed tests were used for all statistical tests. Data distribution was assumed to be normal but this was not formally tested, considering ANOVA is robust to non-normal distributions[48,49]. In instances where the assumptions of parametric tests were potentially violated, we employed more robust alternatives. For multivariate analyses, Pillai's trace was utilized to assess the significance of effects when assumptions of homogeneity of variance-covariance were in question. For $t$-tests, where the assumption of equal variances was challenged, Welch's test was employed. Additionally, in post hoc comparisons, the Games-Howell procedure was utilized to account for unequal variances. These approaches ensured reliable statistical inference across varying data distributions and violation scenarios. To enhance result reliability across diverse statistical methodologies, all null findings were validated through Bayesian analysis.

The statistical analysis was conducted using IBM SPSS Statistics version 28.0.1.1 for frequentist analyses, JASP 0.17.1 for Bayesian analyses, and R for estimating confidence intervals around effect sizes in multivariate analysis. This was achieved by bootstrapping 5000 resamples, calculating the effect size for each resample, and estimating confidence intervals using the percentile interval method (at 0.025 and 0.975) of the bootstrap distribution[50].

For all Bayesian tests conducted in this study, default priors provided by JASP software were utilized. Specifically, for Bayesian $t$-tests, a Cauchy scale parameter of 0.707 was employed. In the case of one-way ANOVA, default values were adopted, including prior specifications on coefficients for fixed and random effects. For coefficient priors, the scale for fixed effects was set to 0.5, while for random effects, it was set to 1.

### Measures

The evaluation of participants' intergroup prejudice, intergroup anxiety, and social engagement was conducted using the measures below.

To measure intergroup prejudice, we used Bogardus' (1925)[51] social distance scale. Participants indicated their willingness on a one to seven scale to be in a social relationship with an outgroup member (i.e., marriage (1), close friend (2), neighbor (3), co-worker (4), citizenship in my country (5), visitor to my country (6), exclude from my country(7))[52–54]. The scale is cumulative, meaning that selecting a lower number, such as 1 (marriage), implies agreement to all more distant relationships. Thus, participants selected only one option that best represented their level of acceptance. Lower scores on this scale indicate a shorter social distance, reflecting a greater willingness to be in a relationship, while higher scores indicate a longer social distance, reflecting a lesser willingness[55]. Outgroup here refers to the ethnicity of the outgroup members involved in the discussion. For example, for a Pashtun participant who was selected to participate in a discussion group comprising Pashtuns and Hazaras, the questions asked their willingness to be in a social relationship with someone from the Hazara ethnic group and vice versa.

To measure intergroup anxiety, we used Swart et al.'s[56] adapted version of Stephan & Stephan's (1985)[57] Intergroup Anxiety Scale, comprising a 6-item bipolar-adjective scale. The adapted version was chosen to simplify the language requirements for our participants who

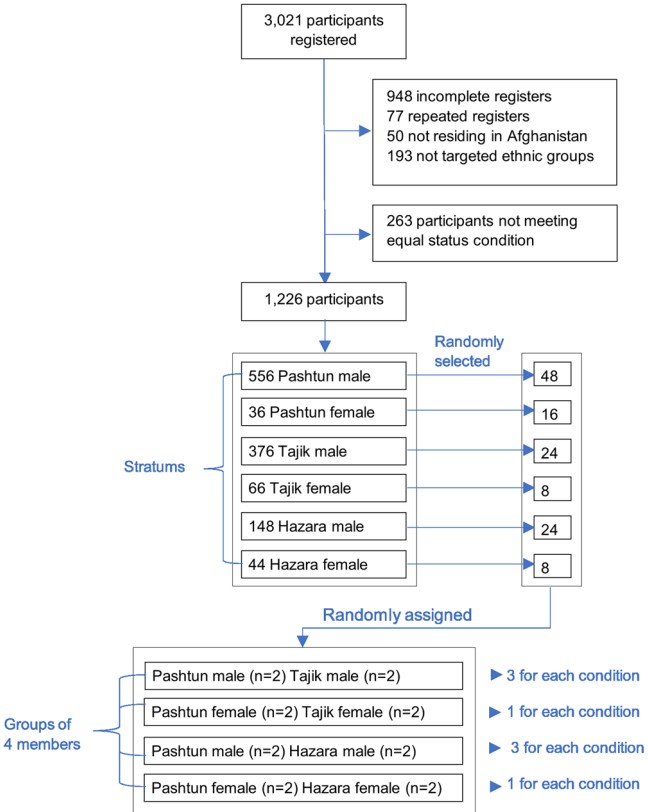

**Fig. 1 | Flowchart of participant recruitment and assignment to study groups.** The flowchart depicts the total number of participants registered (3021), reasons for participant exclusion, and the selection criteria applied to meet the equal status condition of Allport's theory. It also illustrates the final number of participants selected (1226) and their distribution across ethnicities and sexes, with stratification by ethnicity and sex. Participants are randomly assigned to control (no Conversational Agent facilitation) and experimental conditions (normal CA facilitation, CA prompting for issues, and CA prompting for ideas).

**Table 1 | Demographic information of participants selected through stratified random sampling (n = 128)**

|  | Age (*n*, %) |  |  | Education (*n*, %) |  |  |
|---|---|---|---|---|---|---|
| Pashtun Female | 23–27 | 12 | 75.00 | Bachelor | 12 | 75.00 |
|  | 28–32 | 4 | 25.00 | Master's | 4 | 25.00 |
|  | 33–37 | 0 | 0.00 |  |  |  |
| Pashtun Male | 23–27 | 22 | 45.83 | Bachelor | 36 | 75.00 |
|  | 28–32 | 18 | 37.50 | Master's | 12 | 25.00 |
|  | 33–37 | 8 | 16.67 |  |  |  |
| Tajik Female | 23–27 | 4 | 50.00 | Bachelor | 5 | 62.50 |
|  | 28–32 | 2 | 25.00 | Master's | 3 | 37.50 |
|  | 33–37 | 2 | 25.00 |  |  |  |
| Tajik Male | 23–27 | 13 | 54.17 | Bachelor | 17 | 70.83 |
|  | 28–32 | 8 | 33.33 | Master's | 7 | 29.17 |
|  | 33–37 | 3 | 12.50 |  |  |  |
| Hazara Female | 23–27 | 5 | 62.50 | Bachelor | 7 | 87.50 |
|  | 28–32 | 2 | 25.00 | Master's | 1 | 12.50 |
|  | 33–37 | 1 | 12.50 |  |  |  |
| Hazara Male | 23–27 | 11 | 45.83 | Bachelor | 20 | 83.33 |
|  | 28–32 | 10 | 41.67 | Master's | 4 | 16.67 |
|  | 33–37 | 3 | 12.50 |  |  |  |

were non-native English speakers. Participants were asked how they would feel if they were to participate in an activity or discussion with a group consisting of outgroup members (Hazara, Tajik or Pashtun) where they happen to be the only member of their own group. Participants were asked to rate their emotions, using six sets of bipolar adjectives (scaled from 1–5 with the following anchors: 1 = relaxed, 5 = nervous; 1 = pleased, 5 = worried; 1 = not scared, 5 = scared; 1 = at ease, 5 = awkward; 1 = open, 5 = defensive; and 1 = confident, 5 = unconfident)[56]. The scores were formed by calculating the mean across these six items, with higher means indicating greater intergroup anxiety. Cronbach's α reliability of the measure was 0.790 at Time 1, 0.811 at Time 2, and 0.901 at Time 3.

The assessment of social engagement involved an analysis of participants' engagement patterns, content contributions, and their preferences within the asynchronous discussion format. To determine participants' preferences, a post-discussion questionnaire was administered, asking them to indicate their favored discussion format. Out of 128 respondents, 90 expressed a preference for the three-day asynchronous discussion. Consequently, the subsequent analysis of social engagement was focused on the asynchronous discussion context.

To quantify social engagement, two primary quantitative metrics were employed. First, the count of ideas shared[58,59], categorized based on the IBIS (Issue-Based Information System) classification of D-Agree, was used to gauge the extent of participants' engagement with the discussion content. Second, the length of participants' opinions was measured in terms of word count[60]. These metrics were selected as they could potentially serve as indicators of participants' investment and active involvement in the discussion.

**Study instrument**

We utilized SurveyMonkey, a widely used licensed web-based software, to create online questionnaires for the recruitment, pre-discussion, mid-discussion, and post-discussion surveys. This tool is popular among researchers for its capabilities in designing, distributing, and analyzing survey data[61,62].

E-contact occurred via D-Agree[63], an online text-based discussion support system which provides the option to include automated facilitation (i.e., a conversational agent)[63] for both synchronous and asynchronous discussions. D-Agree is structured based on the Issue-Based Information System (IBIS)[64], a decision support system that facilitates informed discussion[65] and consensus building[66] among users. Users can post new content, interact with other users' content, and express agreement or disagreement using D-Agree's desktop or mobile apps. The platform tracks and organizes users' comments as discussion nodes and labels them as issues (questions requiring resolution), ideas (potential solutions or responses), and arguments (justifications or evidence supporting or refuting positions, with arguments in favor termed 'pros' and those against as 'cons'). This structure aids users in understanding the discussion content and facilitates informed decision-making.

Although, D-Agree can be set to incentivize active participation in discussions through real-time rankings based on earned points for activities such as posting, liking, and replying to opinions[67], we disabled this function to align with the intergroup cooperation condition of contact theory, which emphasizes working together towards common goals without competition.

A CA was used to facilitate and moderate the online discussions in the experimental conditions. The CA was a chatbot integrated in the D-Agree platform that acted in a human-like manner as a facilitator, using a set of predefined rules, patterns and facilitation policies[37]. The CA collected the comments submitted by participants as a tree[68], a hierarchical discussion structure formed by interconnected IBIS elements, using argumentation mining tools[69]. We had three distinct types of facilitation settings for the CA. The first type of CA facilitator (normal CA) was a generalist that facilitated all types of opinions, including issues, ideas and arguments. The second type of facilitator (CA prompting for issues) focused specifically on ideas, pros, and cons, and encouraged participants to share their thoughts on issues related to these categories. The third type of facilitator (CA prompting for ideas) targeted only issues and prompted participants to generate ideas

related to those issues. Here's an example of how the first type of facilitation (normal CA) operated:

*Participant A: How can we promote cultural understanding in a diverse society?*

(The CA, acting as a facilitator, recognizes this post as an 'issue,' appropriately labels it as such, and stores it within the discussion's database.)

*Participant B: I think cultural exchange programs could help.*

(The CA identifies this as an 'idea', linking it to the initial issue raised by Participant A, thereby establishing a structured hierarchy of discussion components.)

*CA: What are the merits of this idea?*

(The CA, adhering to predefined facilitation rules, posts a message to prompt for the merits of Participant B's idea, guiding the conversation.)

*Participant C: I believe one merit is that it can help students appreciate different cultures and foster inclusivity.*

(Participant C responds to the CA's prompt by providing a merit of the idea, thus contributing to the ongoing discussion. CA recognizes this post as a 'pro'.)

D-Agree use argumentation mining methods[69] to automatically extract the IBIS nodes (issues, ideas, pros and cons) provided in the participants' comments and then post targeted messages based on the following procedure:

1. Collect the users' submitted comments on D-Agree[63] as a tree[68].
2. Classify the tree into nodes based on IBIS types[64], using data extraction module classifier called Bidirectional Long Short-Term Memory (BiLSTM)[69].
3. Apply argumentation mining method[69] while targeting any of the four IBIS elements for first type of facilitator, targeting idea, pros and cons oriented nodes for second type of facilitator and targeting issue oriented nodes for third type of facilitator, based on a predefined threshold of 1:3 ratio [agent posting:user submitted comments] during the whole discussion.
4. Apply Natural Language Generation (NLG)[70] to the targeted nodes while posting specific messages for each.
5. Go to step [1] and repeat until discussion ends.

The primary objective of these facilitation types was to investigate the impact of different facilitation styles on intergroup prejudice and anxiety. Specifically, we sought to determine if certain types of facilitation could increase or decrease intergroup prejudice and intergroup anxiety. For instance, the facilitator that constantly raised issues might inadvertently increase intergroup prejudice and anxiety, while the facilitator that prompted for ideas might contribute to intergroup prejudice and anxiety reduction. By examining the effects of each facilitation style on prejudice and anxiety reduction or increase, we aimed to gain a deeper understanding of how facilitation can impact group dynamics and intergroup relationships.

## Procedure

The experiment was conducted in batches over a period spanning from May 23, 2022, to June 9, 2022. The 32 discussion groups were divided into four batches, each consisting of 8 groups, with 2 groups dedicated to each experimental condition. This staggered approach was employed to mitigate potential effects of day-specific shocks on participants' behavior, as described by Condra and Linardi[24].

The procedure began with an initial phase where selected participants were contacted individually by phone. During this stage, they were briefed about the experiment's procedures, compensation, and the confidentiality of the research. Additionally, participants received training on how to use the D-Agree platform. For groups with the CA, participants were also informed about its presence and role in guiding discussions. While they were informed about the general purpose and nature of the study, the specific research hypothesis was not revealed to prevent potential bias.

Prior to entering the discussion phase, participants completed a pre-discussion questionnaire. This questionnaire, filled out online via Survey-Monkey, included a consent form, demographic information, and measures of intergroup prejudice and intergroup anxiety.

Following the questionnaire, participants engaged in a two-hour synchronous discussion using D-Agree. Before this discussion began, discussion spaces were established on the platform, and participants were registered, with the discussion theme and duration set. This discussion required participants to collaboratively prepare a proposal regarding the frozen assets of Afghanistan, a topic chosen for its relevance and familiarity with the participants.

After this synchronous discussion, participants were directed to complete a post-discussion survey. This survey, similar to the pre-discussion one, included a consent form and measures of intergroup prejudice and intergroup anxiety, alongside new questions related to their discussion experiences on D-Agree.

The procedure then moved to an asynchronous discussion phase the following day. Participants were given links to join this discussion, which they could access at any time over three days. The task involved collaboratively preparing a draft policy proposal for upgrading informal settlements. The participants were assigned specific roles in summarizing the team's opinions and the overall proposal.

Concluding the procedure, after the three-day asynchronous discussion, participants were asked to complete a final post-discussion survey. This survey mirrored the previous ones in terms of consent and measures of intergroup prejudice and intergroup anxiety but also included new questions pertaining to the participants' experiences in the second round of discussions.

We implemented several measures to prevent deception and ensure transparency in participant interactions. These measures aimed to provide participants with clear information about the presence and role of the CA, as well as assuring the confidentiality and anonymity of the discussions as described in detail below.

During online discussions, participants were organized into groups of four, and their identities were anonymized while indicating their respective ethnicities. Each participant was represented by a title (Mr./Ms.) followed by letters A to D with their ethnic group indicated, such as Mr./Ms. A/B/C/D Hazara/Pashtun/Tajik and was assigned a smiley face icon of a specific color. They were asked not to change their profile pictures or names during the discussion, which they complied with. The CA's profile had no titles or assigned letters. Its profile name was explicitly set as "AI facilitator". Importantly, the CA's profile picture prominently displayed the term "AI", clearly indicating its artificial intelligence nature. It is worth noting that we intentionally designed the CA to not represent any specific ethnic group, emphasizing its role as a neutral authority figure. Additionally, participants' educational level, proficiency in English, and internet access were taken into account during the recruitment process. These factors contributed to their ability to recognize the CA as a non-human entity.

Furthermore, in line with Allport's intergroup cooperation principle and to clarify the CA's role, we designed a task allocation system to promote collaboration. The theme description explicitly stated that the team comprised four participants, excluding the CA. In the three-day discussions, participants were assigned specific days for summarizing their teams' opinions. For example, on day one, Mr. A was assigned the task of summarizing his/her team's opinions, and on subsequent days, other participants were similarly assigned. The CA's role was distinct. While participants shared opinions, the CA focused on facilitating discussions, using the Issue-Based Information System (IBIS) framework[107]. This approach ensured that participants' opinions were guided without introducing the CA's views. This was to ensure transparency and minimize confusion about the CA's role. Participants easily distinguished their own opinions from those guided by the AI, thereby affirming that it was an automated system designed to enhance discussions. These measures prevented deception, promoted transparency and ensured that participants were aware of the CA's presence throughout the study.

### Ethics & inclusion statement

The authors affirm their commitment to ethical research practices and inclusion considerations throughout the study. The research team, comprising Afghan nationals, actively participated in all phases of the research process, encompassing study design, implementation, discussions on data ownership, considerations regarding intellectual property, and authorship of publications. While there were no direct local contributors, certain services, such as participant recruitment, were outsourced to a professional agency located in Afghanistan. Ensuring local relevance was paramount, and measures were implemented to align the study with the socio-political context of Afghanistan.

Roles and responsibilities among collaborators were clearly defined ahead of the research. Due to the non-recognition of the existing government by the country where the research institute is based, seeking approval from a local ethics review committee in Afghanistan was not feasible. Our study was designed to prevent stigmatization, incrimination, discrimination, or personal risk to participants, and provisions were implemented to prioritize the safety and well-being of all involved. The study received ethics approval from the Ethics Committee of the Graduate School of Informatics, Kyoto University (KUIS-EAR-2021-020). Informed consent was obtained from all participants at three key points during the research process, as detailed in the Procedure subsection above.

The primary objective of this publication is to contribute to the enhancement of ethnic relations in Afghanistan, a commitment evident throughout the study. Our citations encompass a diverse range of local and regional research relevant to our study.

Prior to commencing any research activities, participants received comprehensive information about the study's procedures, including details of the research team and the affiliated institution. At the debriefing phase, participants were informed about the study's overarching goal of promoting positive intergroup interactions among different ethnic groups.

The study was not pre-registered due to uncertainties surrounding our ability to conduct research in Afghanistan, given the sensitive nature of ethnic issues and the unpredictable research environment under the Taliban government.

### Reporting summary

Further information on research design is available in the Nature Portfolio Reporting Summary linked to this article.

## Results

### Ethnic baseline comparison

To test the differences in respondents' intergroup prejudice and intergroup anxiety, a multivariate analysis of variance (MANOVA) was conducted with respondent's ethnicity as the independent factor. The analysis revealed that scores for the combined dependent variables were not significantly different among the three ethnic groups, Wilks' $\Lambda = 0.949$, $F(4, 248) = 1.652$, $p = 0.162$, partial $\eta^2 = 0.026$, CI [0.007, 0.100].

Post-hoc comparisons conducted using the Least Significant Difference (LSD) method for intergroup bias and the Games-Howell procedure for intergroup anxiety, did not show any significant pairwise differences on both dependent variables between Pashtun and Tajik ($p = 0.699$, Cohen's $d = -0.084$, 95% CI [−0.609, 0.442], $BF_{10} = 0.243$ for intergroup prejudice; $p = 0.123$, Cohen's $d = -0.425$, 95% CI [−0.954, 0.105], $BF_{10} = 1.401$ for intergroup anxiety), Pashtun and Hazara ($p = 0.637$ Cohen's $d = -0.102$, 95% CI [−0.628, 0.423], $BF_{10} = 0.249$ for intergroup prejudice; $p = 0.118$, Cohen's $d = -0.464$, 95% CI [−0.995, 0.066], $BF_{10} = 1.679$ for intergroup anxiety) and Tajik and Hazara ($p = 0.941$, Cohen's $d = -0.019$, 95% CI [−0.625, 0.588], $BF_{10} = 0.256$ for intergroup prejudice; $p = 0.988$, Cohen's $d = -0.040$, 95% CI [−0.646, 0.567], $BF_{10} = 0.258$ for intergroup anxiety). The Bayes Factors for intergroup anxiety between Pashtun and Tajik and Pashtun and Hazara fall within the inconclusive range (0.3–3.0), indicating no credible evidence for a difference and no decisive evidence for a lack of difference. The Bayes Factors for intergroup anxiety between Tajik and

**Table 2 | Descriptive statistics of study variables across conditions and time points**

| | CA facilitation condition ($n = 96$) | | | Control condition ($n = 32$) | | |
|---|---|---|---|---|---|---|
| | T1 | T2 | T3 | T1 | T2 | T3 |
| | M (SD) | M (SD) | M (SD) | M (SD) | M (SD) | M (SD) |
| Intergroup prejudice | 3.04 (1.75) | 2.94 (1.66) | 2.78 (1.56) | 2.50 (1.32) | 2.84 (1.39) | 3.00 (1.34) |
| Intergroup anxiety | 1.34 (0.58) | 1.22 (0.48) | 1.17 (0.41) | 1.20 (0.36) | 1.24 (0.41) | 1.20 (0.46) |

Hazara and for intergroup prejudice across all groups indicate moderate support for the null hypothesis.

### Manipulation check for the impact of CA facilitation on social engagement

Statistical analysis, employing independent samples t-tests, confirmed that participants in the treatment group, where a CA facilitated discussions, exhibited a significantly higher degree of social engagement in terms of content contribution compared to those in the control group.

For the number of ideas generated, the treatment group (M = 26.28, SD = 15.54) showed a statistically significant advantage over the control group (M = 19.81, SD = 14.008), $t(126) = 2.088$, $p = 0.039$, Cohen's $d = 0.426$, 95% CI [0.022, 0.829]. Similarly, for the length of opinions (word count), the treatment group (M = 1908.56, SD = 956.150) significantly outperformed the control group (M = 1503.69, SD = 829.441), $t(126) = 2.141$, $p = 0.034$, Cohen's $d = 0.437$, 95% CI [0.032, 0.840].

### CA facilitation reduces intergroup prejudice and anxiety

To examine hypothesis 1, we conducted repeated measures ANOVAs with a $2 \times 3$ design (Condition [CA facilitation, no CA facilitation] × Time [1–3]). The between-subject factor was Condition, and Time served as the within-subject factor. The analyses revealed that for intergroup prejudice, the Condition by Time interaction was significant, Wilks' $\Lambda = 0.932$, $F(2, 125) = 4.560$, $p = 0.012$, partial $\eta^2 = 0.068$, CI [0.018, 0.189]. Pairwise comparisons using paired samples $t$-tests revealed that participants in the CA condition reported a significant decrease in intergroup prejudice between T1 and T3 ($t(95) = 2.129$, $p = 0.036$, Cohen's $d = 0.217$, 95% CI [0.014, 0.419]), while there was a slight increase (see Table 2 for means) with no evidence for a statistically significant difference for the no CA setting condition in intergroup prejudice between T1 and T3($t(31) = -1.882$, $p = 0.069$ Cohen's $d = -0.333$, 95% CI [−0.686, 0.026], $BF_{10} = 0.903$). For intergroup anxiety, there was also a significant Condition × Time interaction, Pillai's Trace = 0.050, $F(2, 125) = 3.309$, $p = 0.040$, partial $\eta^2 = 0.050$, CI [0.012, 0.144]. Participants in the CA condition reported a significant decrease in intergroup anxiety between T1 and T2 ($t(95) = 3.665$, $p < 0.001$, Cohen's $d = 0.374$, 95% CI [0.166, 0.580]) and between T1 and T3 ($t(95) = 3.729$, $p < 0.001$, Cohen's $d = 0.381$, 95% CI [0.172, 0.587]). In contrast, there was no evidence for a statistically significant difference in intergroup anxiety for the no CA condition between T1 and T2 ($t(31) = -0.770$, $p = 0.447$, Cohen's $d = -0.136$, 95% CI [−0.483, 0.213], $BF_{10} = 0.248$), or between T1 and T3 ($t(31) = 0.070$, $p = 0.944$, Cohen's $d = -0.012$, 95% CI [−0.350, 0.326], $BF_{10} = 0.189$). Means and standard deviations of intergroup prejudice and intergroup anxiety at three time points for each condition are presented in Table 2, while Fig. 2 shows their distribution across conditions and time points using violin plots.

### CA facilitation style comparison

To test the influence of the CA facilitation style, $3 \times 3$ (CA facilitation condition [normal CA, CA prompting for issues and CA prompting for

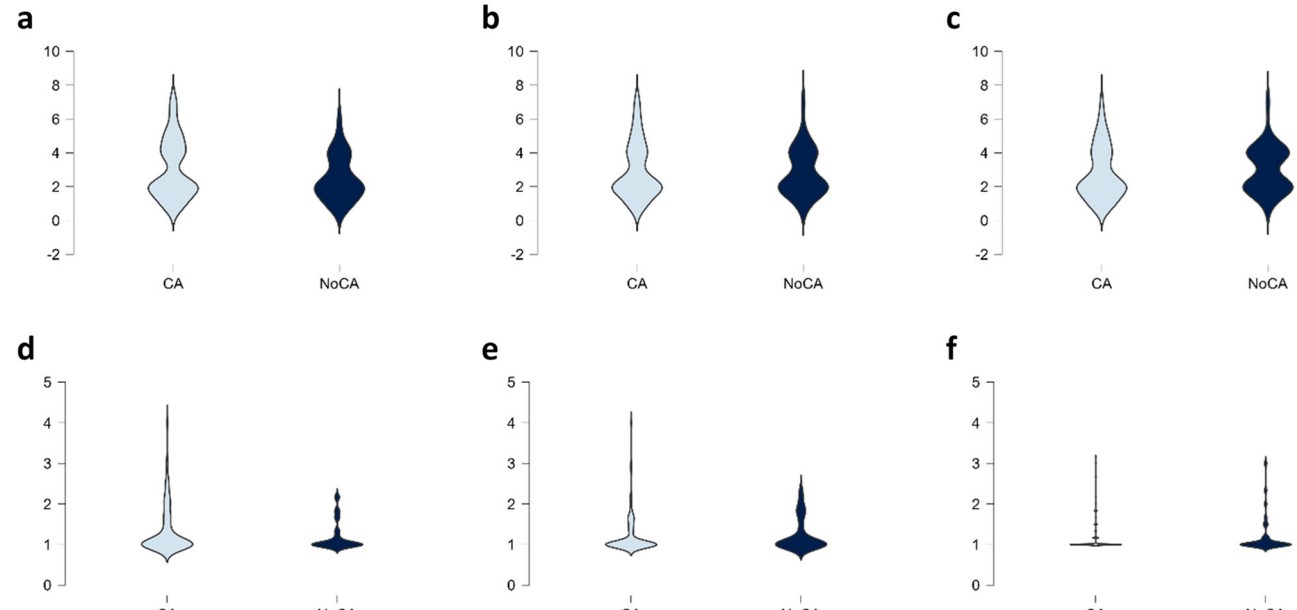

**Fig. 2 | Distribution of intergroup prejudice and intergroup anxiety across conditions (Conversational Agent (CA) facilitation and no CA facilitation) and time points (T1, T2, T3).** Violin plots in panels (**a**, **b**, **c**) represent intergroup prejudice at T1, T2, and T3 for CA facilitation ($n = 96$) and no CA facilitation ($n = 32$). Violin plots in panels (**d**, **e**, **f**) illustrate intergroup anxiety at T1, T2, and T3 for CA facilitation ($n = 96$) and no CA facilitation ($n = 32$). CA conversational agent facilitation condition (experimental), NoCA No conversational agent facilitation condition (control).

ideas] × Time [1–3]) repeated measures ANOVAS were performed. CA facilitation condition was the between-subject factor, and Time was the within subject factor. No statistically significant evidence was found for the Condition X Time interaction for intergroup prejudice, Pillai's Trace = 0.009, $F(4, 186) = 0.206$, $p = 0.935$, partial $\eta^2 = 0.015$, CI [0.010, 0.103], $BF_{Incl} = 0.007$ nor for intergroup anxiety, Pillai's Trace = 0.020, $F(4, 186) = 0.468$, $p = 0.759$, partial $\eta^2 = 0.010$, CI [0.008, 0.086] $BF_{Incl} = 0.027$.

## Discussion
### Research findings and implications
This study contributes to the field of intergroup relations by demonstrating the effective use of a CA to facilitate intergroup E-contact/interactions and mitigate prejudicial attitudes toward outgroups, particularly during a period marked by heightened ethnic tensions. The research findings provide support for our hypothesis, underscoring the role that CAs could play in promoting intergroup interactions, especially in volatile intergroup contexts[20,24,33,34].

Notably, our study's small to medium effect sizes hold particular significance when contextualized within Afghanistan's unique environment, which is characterized by ongoing conflict. This context has been previously associated with heightened prejudice[33] or inconclusive findings[34] in intergroup research. Our study's effect sizes compare to those observed in broader studies, including Pettigrew's influential 2006[15] meta-analysis (mean rs range of 0.205 to 0.214, characterized as small to medium medium) and Imperato, Schneider et al.'s[29] comprehensive 2021 meta-analysis of 23 E-contact studies ($d = 0.36$, characterized as medium). This reaffirms the potential of CA-facilitated intergroup E-contact interventions to yield meaningful shifts in intergroup attitudes.

In addition to confirming the effectiveness of our CA intervention in reducing intergroup prejudice and anxiety, we conducted a manipulation check to assess the broader impact of CA facilitation on participant engagement within the online discussion environment. This assessment included number of ideas and length of contributions which were both greater in the CA versus control condition.

This amplified engagement, while displaying moderate effect sizes, holds important implications[71]. It underscores the role of the CA in stimulating meaningful participation among participants, aligning with broader objectives such as fostering positive intergroup interactions[72] and cultivating more inclusive online discussions. Moreover, this enhanced engagement found in CA-facilitated discussions suggests a potential link to Allport's theory[72], specifically the concept of authority support[9]. Authority support emphasizes the role of institutional or organizational endorsement in nurturing favorable intergroup interactions[12,72–74]. In our study, the CA's explicit association with the research institution positions it as a representative of the organization, possibly leading participants to perceive the CA as an authority figure connected to the research institution[12,72–74]. Future research is needed to confirm if authority support is the mechanism by which CAs lead to greater engagement and better intergroup outcomes.

Furthermore, our study found no evidence for statistically significant differences in intergroup prejudice and anxiety reduction across various CA facilitation policies, which were content-driven and based on the IBIS framework[63,64]. Even when the CA's primary role was to raise issues within participants' opinions based on the IBIS framework, it still played a role in facilitating positive intergroup interactions. In our study, the CA's presence as a non-judgmental moderator[75], regardless of the settings, was associated with better intergroup outcomes.

Our study has practical implications for online discussions and potential internet policing strategies[76,77]. This approach could offer a relatively non-costly way to reduce prejudice online, particularly in situations where direct enforcement might be challenging[76], although future research is needed to demonstrate the feasibility of this approach outside controlled research situations. Given the increased role of the internet in fostering interactions among otherwise physically and socially segregated groups[12,26], this finding is both timely and relevant within the research context and other post-conflict contexts. The fast pace of development in AI/CAs further underscores the importance of this topic.

In closing, this research not only affirms the effectiveness of CA interventions but also offers insights into the interplay between technology, social engagement, and intergroup relations. By shedding light on these dynamics, this study lays the groundwork for designing interventions that encourage less superficial interactions, ultimately leading to reduced prejudice and better intergroup understanding. Researchers should explore various factors influencing online interactions and their potential

effects on intergroup relations, taking into account the evolving landscape of conversational agents.

## Scope and generalizability of findings

The scope conditions of this study revolve around the unique context of Afghanistan, a region that has received relatively little attention in intergroup contact research despite being a country with a history of volatile intergroup relationships[31,33,34,78]. Investigating the effectiveness of indirect contact mechanisms for reducing prejudice in such a complex and conflict-ridden environment[79] is pivotal for several reasons. Firstly, Afghanistan serves as a valuable test case to challenge the traditional boundaries of intergroup contact theories, moving beyond Western, relatively stable contexts. The study contributes to expanding the literature on intergroup contact by demonstrating the adaptability and applicability of contact-based interventions in non-Western, conflict-affected settings. The findings from Afghanistan have the potential to inform similar interventions in regions confronting analogous challenges.

Importantly, while the primary focus of this study was on evaluating the intervention's effectiveness in the Afghan context, it does not specifically predict the ease of applying the intervention in a Western context or in regions without active conflict. Instead, the study provides insights into the effects of the intervention in a conflict-afflicted setting[33,34]. Nonetheless, these findings offer broader implications that may apply to contexts marked by intergroup tensions, encompassing both conflict-affected regions and Western societies during periods of heightened tension[80-82]. The study's insights may prove valuable in addressing prejudice towards various groups in a range of scenarios. For example, they could be relevant in mitigating bias during times of social or political unrest, when tensions between ethnic, racial, or religious groups run high[81,82]. Additionally, the findings may offer insights into reducing prejudice in situations such as online hate speech targeting marginalized communities[80,83] or when dealing with post-conflict reconciliation efforts. Therefore, while this study was conducted in a specific context, its findings offer valuable insights with potential applicability to a diverse array of scenarios characterized by intergroup tensions.

## Ethical considerations in CA-facilitated intergroup contact

This particular section is dedicated to highlighting the ethical considerations that must be taken into account when employing CAs in intergroup contact scenarios. In our study, we ascribe to the CA the role of authority support in line with Allport's theory. However, it is crucial to emphasize that within this context, the CA primarily acts as a facilitator[63], contributing to the organization of online discussions and the cultivation of positive interactions, as previously observed in studies like Kim, Eun et al. (2020)[84] and others[37]. It is also important to clarify that CAs do not possess the authority to monitor or enforce group behavior. This characteristic aligns with the notion that they function as "weak institutions," lacking direct enforcement powers. This dual role of CAs in maintaining structure without enforcing behavior raises substantial ethical considerations. These considerations encompass participant awareness, potential adverse consequences, biases, and the influence of social desirability, all of which warrant examination.

The use of a CA facilitator in our study was motivated by several factors, including scalability[63], consistency, standardized and non-judgmental facilitation[75]. However, this choice raises critical questions about the implications of employing a CA in place of a human facilitator. While CAs excel in impartiality and can efficiently manage interactions on a larger scale, human facilitators possess qualities such as empathy and adaptability, potentially resulting in distinctive participant experiences[85]. It is essential for future research to delve deeper into these distinctions through comparative studies that explore how human and CA facilitators influence participant outcomes in intergroup contact. These investigations can provide valuable insights into the ethical and practical considerations surrounding the choice of facilitation method, thus enriching our understanding of their respective roles in shaping intergroup interactions.

Furthermore, the utilization of a CA facilitator in this study highlights significant ethical considerations demanding thorough examination, particularly pertaining to participant awareness, potential adverse repercussions, biases, and the impact of social desirability[86]. Ensuring that participants are aware of their interaction with a computer program, rather than a human, is of paramount importance. While we took specific measures in this study to mitigate the potential for deception[87], it is imperative for future research to transparently convey the nature of the facilitator to minimize the likelihood of participants mistakenly perceiving the CA facilitator as a human participant. The absence of such awareness could lead to unforeseen consequences, potentially eroding trust and cooperation[88]. Moreover, it is essential to recognize that the level of transparency should be carefully managed and controlled in experiments to ensure that it aligns with research objectives and guide against unintended biases or influences on participant behavior that do not accurately represent real-world scenarios. This consideration is particularly important when studying the effects of CAs on human behavior and attitudes.

Recognizing that AI systems have the potential to perpetuate societal biases is a crucial consideration[89]. This realistic concern can influence study outcomes and the future application of this tool. To address this, we emphasize the necessity of continuous monitoring and evaluation of the CA[90]. This ongoing assessment should scrutinize the CA's responses, prompts, and interactions to detect and mitigate the risk of reinforcing prejudiced attitudes or behaviors. In our study, the CA's facilitation settings were meticulously structured to minimize bias, with AI-generated prompts and messages being content-driven to avoid preferential treatment based on ethnic backgrounds. However, it is essential to acknowledge that while the CA categorizes opinions, it does not inherently recognize biased content within discussions. To comprehensively address this concern in future studies, we recommend (a) exploring the feasibility of enhancing the CA's ability to recognize and address biased content, particularly in scenarios where participants express prejudiced viewpoints, which might necessitate refining algorithms to identify biased language or ideas, while also ensuring that the CA's responses remain unbiased and impartial, and (b) implementing reporting mechanisms within the discussion platform for participants to report instances of perceived bias or biased content. Such mechanisms can be invaluable in identifying and rectifying bias-related issues promptly.

Moreover, concerns related to social desirability bias may emerge, where participants might feel compelled to provide socially desirable opinions due to the awareness of being observed (the Hawthorne effect[91]) by a CA. This pressure can impact the authenticity of participant contributions, potentially casting doubt on study findings. To address this concern, our study leveraged CA facilitation to ensure a supportive and non-judgmental environment, which actively encouraged participants to express their genuine thoughts and opinions.

In conclusion, this study was conducted with meticulous attention to ethical considerations. Stringent measures were implemented to prevent deception and ensure transparency, which encompassed participant awareness, potential consequences, biases, and the influence of social desirability[92]. Ongoing evaluations of the facilitator bot and its impact on intergroup cooperation are pivotal in addressing ethical concerns to guide future implementations.

## Limitations

This study has several limitations that should be acknowledged. First, in terms of sample size, we had four equal-sized groups for the four types of settings of the CA: (1) no CA (2) the normal CA (3) CA prompting for issues (4) CA prompting for ideas. We compared the control group with the combined participants of all CA groups, which led to unequal sample sizes. The difference in sample size is a limitation to consider when interpreting the results. Future studies should strive to conduct well-powered studies with equal sample sizes across conditions to enhance the reliability and generalizability of the findings.

Second, our participants' education, fluency in English, and access to the internet suggest that they may come from a higher socioeconomic background, which may not accurately represent the views of the average

Afghan citizen. However, this was done to ensure equal status of the contact participants, which is one of the optimal contact effectiveness prerequisites suggested by Allport (1954)[7].

Third, in this study, we introduced CAs as part of the research design to facilitate intergroup E-contact. However, it is important to acknowledge that in real-world scenarios, the adoption of CAs might not occur naturally. The controlled experimental setting of our study allowed us to assess the effects of CAs under specific conditions, but the translation of these findings to real-life situations should be approached with caution.

## Conclusion

In summary, this study showed that using a conversational agent (CA) in intergroup electronic contact (E-contact) can reduce interethnic prejudices and hostility among Afghanistan's historically segregated ethnic groups. The CA led to increased participation, longer discussions, and a greater reduction in outgroup prejudice and anxiety compared to the control condition, with moderate effect sizes similar to broader meta-analyses on intergroup contact. The findings support the hypothesis that CAs can promote positive intergroup interactions and underscore the importance of considering ethical implications, such as transparency and bias mitigation, when using CAs in intergroup contact scenarios. Despite limitations like unequal sample sizes and a controlled setting, the study provides valuable insights into the mechanisms that may help to reduce prejudice in conflict-affected settings and suggests potential for broader applicability of CA-facilitated interventions in diverse intergroup contexts.

## Data availability

The dataset that underlies Table 2 is deposited and can be found at the link below. https://osf.io/ydnwz/?view_only=9661e2571fae4ac6999ba129bf8d726a. Other data that support the findings of this study are not openly available due to reasons of sensitivity. The data are available from the corresponding author upon request. Requests for access will be reviewed within 30 days by the corresponding author. The data can be used only for academic research purposes via a data use agreement. Access, if granted, will be provided in a manner consistent with the original informed consent and privacy assurances given to study participants.

## Code availability

All code used for statistical analyses is available online at the link below. https://osf.io/js5dq/?view_only=838cecb22e1840f39b059a45bfb563dc.

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

## Acknowledgements

This research was supported partially by the JST CREST fund (Grant Number: JPMJCR20D1, Japan) and JSPS KAKENHI (Grant Number: 22K17948, Japan). The funders had no role in study design, data collection and analysis, decision to publish or preparation of the manuscript.

## Author contributions

The study was conceived and designed by S.S., with feedback provided by the other authors. The experiment was designed by S.S. with input from R.H., S.OH., J.H. and T.I. Subject selection and randomization were carried out by S.S. assisted by R.H. The experiment was conducted by J.H and S.S. Data analysis was performed by S.S. The manuscript was written by S.S., with guidance on paper structure, writing specific sections, and revisions provided by R.E.I. In addition, J.H. contributed to the methods section by writing on the study instrument. The implementation of the types of facilitation policies to the conversational agent was carried out by T.H. All authors provided intellectual inputs into aspects of this study and approved the final version.

## Competing interests

The authors declare the following competing interests: T.I. served as the Principal Investigator (PI) for the development of the online discussion platform utilized in this research, part of the "Innovating Agent-based Large-scale Consensus Support System" project. This project, supported by the Japan Science and Technology Agency (JST) under the CREST program (Grant No. JPMJCR15EI), received funding from October 2015 to March 2021. The intellectual property (IP) rights of the online platform are jointly owned by Nagoya Institute of Technology and Kyoto University, both national research universities in Japan, and all associated activities within the university are focused on nonprofit research and development. T.I. is also one of the inventors for certain aspects of the technology, for which a United States patent application has been filed (application no: PCT/JP2019/031183, Publication no: US 2021/0319187 A1). All other authors have no competing interests to declare.

## Additional information

**Supplementary information** The online version contains Supplementary Material available at https://doi.org/10.1038/s44271-024-00070-z.

