## [Peer Review File · Communications Psychology]

22nd May 23

Dear Dr Sahab,

Thank you for your patience during the peer-review process. Your manuscript titled "Reducing Interethnic Bias with Conversational Agent in E-contact Experiments in Afghanistan" has now been seen by 3 reviewers, whose comments are appended below. You will see that they find your work of some potential interest. However, they have raised quite substantial concerns that must be addressed. In light of these comments, we cannot accept the manuscript for publication, but would be interested in considering a revised version that fully addresses these serious concerns.

We hope you will find the Reviewers' comments useful as you decide how to proceed. Should additional work allow you to address these criticisms, we would be happy to look at a substantially revised manuscript. If you choose to take up this option, please highlight all changes in the manuscript text file, and provide a detailed point-by-point reply to the reviewers.

Editorially, we consider it critical that you engage in a full revision of the manuscript so that it addresses the reviewers' concerns. This includes a revision of the introduction (i.e., early introduction of the research question and contribution of the manuscript, theoretical conceptualization of the study drawing on recent literature, contextualization of the Afghanistan setting, and a presentation of the hypotheses that accurately corresponds to your research question; we support the referee's request to renumber the hypotheses (alternative hypotheses to the H1 should be presented as such). You must also rewrite the Methods and Results sections opting for greater clarity and transparency. All relevant information and analysis should be in the main manuscript. You may use up to 10 display items, and extended data items are not permitted.

As you revise the Results section, please make sure to add all additional requested information and report full statistical results including effect sizes. Please also ensure that all information contained in the Reporting Summary is redundantly included in the Methods/Results. When omnibus tests are not significant, there is no need for further decomposition of the findings. Please ensure to use appropriate language to describe any null results. (There is no statistical test that can demonstrate absence of an effect. Statements such as 'There is no difference between x and y.' or 'X does not affect Y.' must be revised to read 'We found [no/little] credible evidence of a difference between x and y.' or 'We found [no/little] credible evidence that X affects Y.').

For the general structure, please ensure the formatting of your manuscript follows these guidelines: Communications Psychology formatting guide. Finally, please include an ethics and inclusion statement, as indicated in the Editorial Policy checklist (please follow the links in the checklist for further information) and include information about the patent and potential conflicts of interest in the manuscript file.

We ask that custom analysis code is deposited; please note that this pertains to the SPSS script you use to analyse your data. You will find more information on our Code sharing requirements below. We appreciate that you note restrictions on data sharing, but please ensure that the description of

limitations to access etc. follows our guidelines (more below under Data Availability). Please also note that the fully anonymized numerical data underlying charts and plots must be deposited publicly at the time of publication. We strongly recommend taking this step now.

If the revision process takes significantly longer than five months, we will be happy to reconsider your paper at a later date, provided it still presents a significant contribution to the literature at that stage.

Please use the following link to submit your revised manuscript, point-by-point response to the Reviewers' comments with a list of your changes to the manuscript text (which should be in a separate document to any cover letter) and any completed checklist:

[link redacted]

Please do not hesitate to contact me if you have any questions or would like to discuss the required revisions further. Thank you for the opportunity to review your work.

Best regards,

Jennifer Bellingtier

Jennifer Bellingtier, PhD
Senior Editor
Communications Psychology

EDITORIAL POLICIES AND FORMATTING

Editorial Policy: Policy requirements (Download the link to your computer as a PDF.)

Furthermore, please align your manuscript with our format requirements, which are summarized on

the following checklist:

Communications Psychology formatting checklist

and also in our style and formatting guide Communications Psychology formatting guide .

* **CODE AVAILABILITY:** All Communications Psychology manuscripts must include a section titled "Code Availability" at the end of the methods section. In the event of publication, we require that the custom analysis code supporting your conclusions is made available in a publicly accessible repository; please choose a repository that provides a DOI for the code; the link to the repository and the DOI must be included in the Code Availability statement. Publication as Supplementary Information will not suffice. We ask you to prepare and upload code at this stage, to avoid delays later on in the process.

* **DATA AVAILABILITY:**

All Communications Psychology research manuscripts must include a section titled "Data Availability" at the end of the Methods section or main text (if no Methods). More information on this policy, is available at <http://www.nature.com/authors/policies/data/data-availability-statements-data-citations.pdf>.

At a minimum the Data availability statement must explain how the data can be obtained and whether there are any restrictions on data sharing. Communications Psychology strongly endorses open sharing of data. If you do make your data openly available, please include in the statement:

We recommend submitting the data to discipline-specific, community-recognized repositories, where possible and a list of recommended repositories is provided at <http://www.nature.com/sdata/policies/repositories>.

If a community resource is unavailable, data can be submitted to generalist repositories such as figshare or Dryad Digital Repository. Please provide a unique identifier for the data (for example a DOI or a permanent URL) in the data availability statement, if possible. If the repository does not provide identifiers, we encourage authors to supply the search terms that will return the data. For data that have been obtained from publicly available sources, please provide a URL and the specific data product name in the data availability statement. Data with a DOI should be further cited in the methods reference section.

REVIEWER EXPERTISE:

Reviewer #1 Intergroup contact

Reviewer #2 Intergroup contact

Reviewer #3 Intergroup contact

Reviewer #1 (Remarks to the Author):

In this manuscript the authors set out to investigate the intervening effects of the conversational agent (CA) of computer-mediated (E) inter-ethnic contact on prejudice in a (post)conflict setting of Afghanistan. They found that inter-ethnic E-contact reduced prejudice over time under facilitation by CA. Under the conditions of no facilitation of the CA, E-contact led to an increase of prejudice. There is much to appreciate about this study and in particular the setting where the study was conducted and exploration of the moderating role of CA facilitation of contact effects on prejudice. However, at the same time, there are some major limitations (which could be addressed with follow-up studies) that prevent me from recommending this manuscript for publication. I will list some of my main concerns:

1. The main question of the effects and underlying mechanism of the CA effects remain largely unanswered. While the evidence reported here supports the hypothesis of the CA facilitation leading to prejudice reduction, the issue as to why remains unanswered. Furthermore, the reasoning behind the expected positive intervening effect of the CA remains unclear in the introduction.
2. Even though the authors acknowledge the limitation of unequal sample sizes across conditions, I find this limitation to be crucial for inferring the robustness of already very small effects. I would suggest the authors to report and comments on effect sizes for this study and to ensure well-powered and equal sample sizes in future studies.
3. Finally, the difference and conflation between indirect contact and computer-mediated contact remains inconsistent and unclear.

In conclusion, there is much to appreciate about this work. I applaud the authors on this line of research in such contexts. However, the robustness of reported evidence is limited by some major methodological constraints which I hope the authors will be able to address in the future.

Reviewer #2 (Remarks to the Author):

This article tests an intervention involving indirect intergroup contact (computer-based with a bot facilitator) in an understudied context, Afghanistan. It offers promising findings to this vast literature. The authors also helpfully lay out the many ways we can interpret and more importantly, try to implement, the intergroup contact theory — from comparing direct face-to-face contact with indirect virtual contact, to needing a mix of both.

My main comments are the authors should be more upfront that the conversational agent (CA) in the study context is NOT the outgroup participant but rather a facilitator of e-conversations between real participants — how does that fit into intergroup contact theory, how did that work in reality. They need a theoretical section dedicated to thinking through the implications of this facilitator bot,

particularly focusing on ethical implications and deception. They can also do more to help the reader think through both the possible positive and negative effects of e-contact, as well as discuss concerns around social desirability and scope conditions. Please find more detailed comments below.

Main comments:

Based on how the introduction and theory sections are set up, I thought the CA was pretending to be a person from the other group, and the study was about having people chat with a bot who appeared to be a person from the outgroup. That is not the case. From the hypothesis, we later learn that the CA is merely facilitating the contact. So this study is less about testing the effects of e-contact, but more about comparing a CA-facilitated e-contact versus not facilitated. To bring this into Allport's conditions, it's like changing the condition between whether the contact has received support from some authority figure or not. The authors should be clear about this from the onset.

Relatedly, the authors need to tackle this condition more in the theory. Even in the results and discussion sections, it's unclear why having a bot facilitate the discussion would lead to larger prejudice-reduction effects compared to not having that bot.

Then again, how is the CA (particularly a bot) considered an authority figure? This is not what Allport had meant by institutional and authority support, right? Might the participants interpret this bot as connected to government or an NGO in some way? For future implementation, such as if an organization used this tool, the bot would have to appear as if it was coming from that organization, right?

In the theory section, on pages 4-5, when authors write about voluntary participation and voluntary contact, eg. in the theory section on page 4, are they referring to self-selection, as in, if people self-select into contact with outgroups then they are more likely to feel less prejudiced? If that's the argument, it seems difficult to disentangle what the effects of contact are versus self-selection bias — these people are already less prejudiced and open to contact to begin with.

On page 8, the authors write "E-contact also has the potential to generate unintended consequences," but leave it at that. Can the authors go on to elaborate what those unintended (negative) consequences might be?

It would be helpful to include examples of how the CA facilitated conversations. Much of the Discussion Instrument in the appendix could be in the main manuscript. And it would be helpful to have concrete examples of what these conversations and facilitators looked like.

Using the CA implies deception — were participants aware that they were being facilitated by a computer program, not a real person? Who did they assume the CA was — Pashtun or non-Pashtun?

Typically, this kind of deception brings many negative ethical implications, particularly in the context of the study if indeed prejudice against the outgroup was already so heightened. There could certainly be negative consequences if people realize that their facilitator was simply a computer program and not a real person, and any positive effects could vanish. How do the authors grapple with the ethical issues of the study?

The ethics section in the appendix right now just says the study was IRB approved. That is often just the bare minimum in ethical obligations for studying vulnerable populations.

As an implication for this study, what if the CA appears to show bias? We know that AI often reproduces existing societal biases. Could this be a realistic concern for this study, and future possible implementation of this tool?

Was it not a possibility to actually allow for e-contact between participants from both groups, and include a trained enumerator to facilitate? Why use a bot? Or why not compare a human versus bot facilitator?

What about concerns around social desirability? With the facilitator, participants might have felt more pressure to answer in socially desirable ways since they knew they were being observed (Hawthorne effect).

In terms of work that should be cited: see Nejlja Asimovic's research around social media facilitated intergroup contact, Salma Mousa's e-contact research as well as her 2020 Science paper that shows contact with outgroup members may only change attitudes about those specific members not the outgroup as a whole. Lastly, Zhou and Lyall 2023 used an RCT to test a large scale, prolonged contact intervention in Kandahar, Afghanistan and found null effects.

What are the scope conditions of this study? While the authors argue that Afghanistan is a lesser studied context for intergroup contact research, what should we learn from this context specifically? How does this new context add other theoretical or empirical contributions? Do the authors anticipate that their intervention would work more easily in a western context or in a place that was not experiencing active conflict? They should also discuss in greater detail how ongoing violence and political conflict might shape how the intervention is designed and its success?

Minor comments:

In the abstract and intro, describe the magnitude of the effects. Hard to gauge with simply positive and statistically significant.

These sentences were confusing:

On p4: "In other words, the more intimate contacts that members of an ingroup have with members of an outgroup, the more positive their attitudes and relationship with such outgroup and indeed other outgroups not necessarily encountered in the initial contact situation."

Are the authors trying to say that positive contact with a few members of an outgroup can less prejudice against that outgroup as a whole? Or that Allport's conditions don't necessarily need to be met?

On p5: "However, despite the widespread appeal that the contact hypothesis has received in academic and policy circles, some scholars have cautioned against the credulous association of intergroup contact with positive outcomes."

Reviewer #3 (Remarks to the Author):

Reducing Interethnic Bias with Conversational Agent in Contact Experiments in Afghanistan

The authors point out that while much work has been done over the years to examine the contact hypothesis through direct forms of inter-group exposure, less has been done to examine more indirect contact, especially involving new online technologies. Using a multi-stage indirect contact experiment among Afghans, the authors find that conversation agents (CA) on online discussion platforms reduce interethnic prejudice. I believe the manuscript could potentially be published in *Communications Psychology* but it would require major revisions, but given the extent of those revisions, I recommend that the manuscript either be rejected or “rejected and resubmitted”.

1. Clarify the manuscript’s contribution. The authors state in the literature review that “A meta-analysis conducted by Imperato and others (2021) on the effects of virtual platforms on prejudice reduction revealed that like direct contact, contact through online platforms had significant positive effects on the transformation of prejudicial attitudes notwithstanding the different samples involved and the research context.” Identify this manuscript’s contribution in light of existing literature on the topic more clearly. The discussion in lines 232-241 on the manuscript’s contribution should come earlier.
2. Direct vs. Indirect contact discussion. Some of the manuscript’s focus on direct vs. indirect contact effects seems tangential given that the authors are not evaluating direct contact effects.
3. Theorizing positive effects of online indirect contact. Instead, the authors should focus more on the controversy surrounding indirect contact online, and whether it reduces or enhances prejudice. The authors should provide a theoretical framework to explain when indirect contact should work to reduce prejudice online and test hypotheses using the case at hand. Identify key scope conditions in relation to CAs. What do CAs represent? They signal “rules of the game” for online discussion, but they do not have monitoring or enforcement power over group behavior, so they are weak institutions at best. They are also exogenous, and it’s unclear whether groups would adopt CAs on their own in nature. Still, if weak institutions can reduce prejudice online in relatively non-costly ways, then the study has useful, practical policy implications for internet policing.
4. Rationale for Afghanistan. Why study this case? What makes Afghanistan compelling for examining indirect contact mechanisms for reducing prejudice? This needs clarification. How generalizable are the results to other contexts?
5. Hypotheses: H1 and H2 seem the same to me. H3 is the null. Why not just simplify this to one hypothesis on the effectiveness of CAs in reducing prejudice? H2 and H3 are not needed. H4 is not well-theorized and needs further development to understand the causal process of mediation. I think what the authors really argue is that anxiety is a moderator, not a causal mediator.
6. Research Design. The authors need a clear research design section that establishes the experimental design, how it tests the hypotheses and identifies the causal effects of CA. This is currently missing. This makes the Results section all the more difficult to follow. The presentation is more like a mystery novel, where we are focused to infer the design from the discussion of results. Time effects in the design are also not theorized or hypothesized clearly. What is the CA? How is it operationalized? So much is unclear.
7. Results. The presentation of results should focus on testing key hypotheses about CA effects. How are intergroup prejudices measured? Table 1 suggests to me that CAs have a potentially weak reductive effect. Figure 1 seems unnecessary. Why is this even in the manuscript? It doesn’t establish a clear mechanism in relation to CAs on reducing prejudice (or anxiety). The mechanism is unclear. Also, it’s unclear how either of these things are measured empirically.
8. Discussion. Given the preceding problems, it is difficult to draw implications from the study in the

discussion that follows. The section at the end on study limitations raises questions about sample size (not reported in the manuscript), sampling bias (who are the participants?), the consent process (was deception involved? Were respondents debriefed? Did the study have IRB approval? When was it conducted? Ethical implications?). This should come before the discussion section, ideally as part of a section on data collection that precedes the results.

In summary, my initial enthusiasm to review this manuscript based on the abstract has been dampened by disappointment in the execution of the manuscript for the reasons stated above. I could go into other issues, but I trust that other reviewers will raise similar points. My comments should not discourage the authors from revising the manuscript, but the authors should really work on organizing the manuscript before resubmitting it either here or elsewhere. Finally, here are questions that the authors should especially focus on from the editors of CP.

-Does the paper represent an advance in understanding which may influence thinking in the field? I would argue yes potentially.

-Does the article presents an original study, new analysis, new model, or a direct or extended replication of previous work? An original study, clearly.

-Are the data and analysis technically sound? Are they appropriate to answer the research question, e.g., are causal research questions addressed on the basis of causal, rather than correlational evidence? Here there is real room for improvement.

-Does the paper provide strong evidence for its conclusions? The causal effects seem weak at best.

-Is the study question important to scientists for a sub-field of psychology? Potentially

-Are there any special ethical concerns arising from the use of animals or human subjects? Potentially, these are not addressed clearly in the manuscript.

- Was the study preregistered and if so, did the authors follow the preregistration? This is unclear from the manuscript.

Reviewer 1

In this manuscript the authors set out to investigate the intervening effects of the conversational agent (CA) of computer-mediated (E) inter-ethnic contact on prejudice in a (post)conflict setting of Afghanistan. They found that inter-ethnic E-contact reduced prejudice over time under facilitation by CA. Under the conditions of no facilitation of the CA, E-contact led to an increase of prejudice. There is much to appreciate about this study and in particular the setting where the study was conducted and exploration of the moderating role of CA facilitation of contact effects on prejudice. However, at the same time, there are some major limitations (which could be addressed with follow-up studies) that prevent me from recommending this manuscript for publication. I will list some of my main concerns:

Response

We sincerely appreciate your thorough evaluation of our manuscript and valuable feedback. We are also grateful for the recognition of the significance of our research, which investigated the effects of a conversational agent (CA) in computer-mediated inter-ethnic contact on prejudice in the (post)conflict setting of Afghanistan. Your positive remarks regarding the exploration of the moderating role of CA facilitation on contact effects are encouraging. In response to your specific concerns, we have provided a point-by-point response below.

R1 Comment 1.

The main question of the effects and underlying mechanism of the CA effects remain largely unanswered. While the evidence reported here supports the hypothesis of the CA facilitation leading to prejudice reduction, the issue as to why remains unanswered. Furthermore, the reasoning behind the expected positive intervening effect of the CA remains unclear in the introduction.

Response

We thank reviewer 1 for the valuable insight into the need for a more explicit exploration of the effects and underlying mechanisms of the Conversational Agent (CA) on prejudice reduction. In response to this constructive feedback, we have made substantial revisions to shed light on the “why” behind the CA's impact on prejudice reduction.

In our introductory section, “E-Contact and Conversational Agents” (lines 220-270), we have expanded our explanations of the CA's pivotal role in facilitating intergroup E-contact. Specifically, we emphasize the CA's function as a form of authority support, which aligns with Allport's Contact Hypothesis consistent with previous studies (see^{1,2}). Furthermore, we delved into how the CA's presence enhances social presence³ and fosters human-like interaction and intensifying engagement⁴. This heightened engagement significantly influences attitudes and behaviors, as participants feel a stronger connection and familiarity during the discussion. Additionally, we explained how the CA's guidance during discussions can create opportunities for cognitive dissonance⁵, ultimately leading to attitude change.

To substantiate these claims, we conducted statistical analyses that confirm participants in the treatment group, where a CA facilitated discussions, exhibited a significantly higher degree of social presence in terms of content contribution compared to those in the control group. These results are detailed under a new subsection titled “Manipulation check for the impact of CA facilitation on social presence” (lines 616-626) in the “Results” section. We have also discussed this in the “Introduction” lines 81-82 and “Discussion” section (lines 671-687).

These additions offer a more comprehensive explanation of the underlying mechanisms through which the CA is anticipated to contribute to prejudice reduction. Therefore, we believe that our revisions effectively address the reviewers' comments.

R1 Comment 2.

Even though the authors acknowledge the limitation of unequal sample sizes across conditions, I find this limitation to be crucial for inferring the robustness of already very small effects. I would suggest the authors to report and comments on effect sizes for this study and to ensure well-powered and equal sample sizes in future studies.

Response

We appreciate the reviewer's insightful comment regarding the importance of ensuring well-powered and equal sample sizes in research studies. As noted, we have acknowledged the limitation of having unequal sample sizes across conditions, and we understand its potential impact, especially when dealing with small effect sizes.

In response to the reviewer's suggestion, we have incorporated effect sizes, specifically Cohen's d , into the pairwise comparisons (lines 635-636, 639-640), and have commented on these effect sizes in the discussion section lines 662-669.

Our study's small to medium effect sizes hold particular significance when contextualized within Afghanistan's unique environment, which is characterized by ongoing conflict. This context has been previously associated with heightened prejudice⁶ or inconclusive findings⁷ in intergroup research. Furthermore, our study's effect sizes compare to those observed in broader studies, including Pettigrew's⁸ influential 2006 meta-analysis (mean r s range of 0.205 to 0.214, characterized as small to medium medium) and Imperato, Schneider et al.'s⁹ comprehensive meta-analysis of 23 E-contact studies ($d = 0.36$, characterized as medium). This reaffirms the potential of CA-facilitated intergroup E-contact interventions to yield meaningful shifts in intergroup attitudes.

We acknowledge the vital importance of well-powered and equal sample sizes in future research endeavors by ensuring such sample sizes is paramount for enhancing the reliability and generalizability of research findings. To this end, we have incorporated text in the “Research Limitations and Future Research” section (lines 800-801, 816-817) that discusses our commitment to addressing this issue in future studies.

R1 Comment 3

Finally, the difference and conflation between indirect contact and computer-mediated contact remains inconsistent and unclear.

Response

Thank you for your feedback on the difference and conflation between direct and indirect contact. We acknowledge that different reviewers have offered varying perspectives. Reviewer 2 highlighted our contribution regarding indirect intergroup contact and our discussion on different interpretations and implementations of intergroup contact theory “from comparing direct face-to-face contact with indirect virtual contact, to needing a mix of both.” In contrast, Reviewer 3 suggested focusing less on the distinction between direct and indirect contact and delving into the controversy surrounding how indirect contact impact prejudice reduction in online contexts.

To address these varying viewpoints, we have made revisions to provide a balanced discussion that covers both theoretical aspects and controversies related to indirect contact (see lines 154-172). Additionally, we offer further clarification below.

It is subject to debate whether there are differences between indirect contact and computer-mediated contact, as the literature does not make a clear distinction between the two (see¹⁰, p. 2; 11).

However, to the extent that computer-mediated contact makes it possible for participants to conceal many of the physical, cultural and social attributes which would have otherwise been revealed in direct contacts (see¹²), it qualifies to be conceptualized as indirect. Indeed, some of the leading scholars on E-contact assert that: “...E-contact can be considered indirect, in the sense that the contact can be non-FtF (e.g., through text) and is mediated by a computer device or software. Individuals are not required to be in the same physical space when interacting, but rather the same cyber space... E-contact can also be considered a more direct form of contact in that the synchronous nature of the Internet text chat tool ensures individuals interact in real time, allowing for the actual engagement of self in the immediate contact situation.”¹³, p. 131.

Similarly, White & Abu-Rayya¹⁴, p. 598, note that under E-contact, “ingroup and outgroup members never physically meet or see one another, but interact in a text-only fashion using synchronous internet chat tool. Thus, the text-only or non-face-to-face nature of the contact ensures that it remains indirect, however the synchronous nature of the internet chat has the added advantage of the actual engagement of self in the immediate contact situation.” (see also¹).

R1 Comment 4.

In conclusion, there is much to appreciate about this work. I applaud the authors on this line of research in such contexts. However, the robustness of reported evidence is limited by some major methodological constraints which I hope the authors will be able to address in the future.

Response

We sincerely appreciate the reviewer's positive feedback and her/his recognition of the value of our research in challenging contexts. We also acknowledge the methodological constraints mentioned. In response to these concerns, we have already taken several significant steps to strengthen our research, as outlined in our previous response. These steps include:

1. Providing detailed explanations and conducting in-depth analyses to illuminate the mechanisms through which CA facilitation is anticipated to mitigate ethnic prejudice.
2. Clarifying the introductory sections to better underscore the study's contribution to the existing literature.
3. Renumbering the hypotheses for greater clarity.
4. Revising the methods, results, and discussion section to enhance transparency and overall comprehension.
5. Reporting comprehensive statistical results, including effect sizes.
6. Furnishing comprehensive information on the ethical considerations concerning the use of CA and transparently addressing our research limitations.

These actions collectively reinforce the rigor and comprehensiveness of our research. We remain committed towards enhancing the robustness of our research in future studies. We will diligently consider and address these constraints in our upcoming research endeavors to ensure quality and reliability of our findings.

Reviewer 2

This article tests an intervention involving indirect intergroup contact (computer-based with a bot facilitator) in an understudied context, Afghanistan. It offers promising findings to this vast literature. The authors also helpfully lay out the many ways we can interpret and more importantly, try to implement, the intergroup contact theory — from comparing direct face-to-face contact with indirect virtual contact, to needing a mix of both.

R2 comment 1.

My main comments are the authors should be more upfront that the conversational agent (CA) in the study context is NOT the outgroup participant but rather a facilitator of e-conversations between real participants — how does that fit into intergroup contact theory, how did that work in reality. They need a theoretical section dedicated to thinking through the implications of this facilitator bot, particularly focusing on ethical implications and deception. They can also do more to help the reader think through both the possible positive and negative effects of e-contact, as well as discuss concerns around social desirability and scope conditions. Please find more detailed comments below.

Response

We appreciate the reviewer's thoughtful consideration of our paper and her/his recognition of its contributions to the field of intergroup contact theory, particularly within the context of Afghanistan. We also value their perspective on the various interpretations and implementations of this theory, which we aim to enrich through our study.

Regarding the role of the conversational agent (CA) in our study, we acknowledge the importance of clarity in understanding its function. In response to the reviewer's concerns, we have refined our introductory section and discussions to provide a more comprehensive explanation of the CA's role as a facilitator of e-conversations among real participants. This clarification aligns with our commitment to transparency in addressing potential ethical implications and deception, as the reviewer suggested. We have expanded on the ethical considerations related to the CA and the potential positive and negative effects of e-contact in the revised manuscript.

Furthermore, we have incorporated discussions on the scope conditions of our study and addressed concerns related to social desirability. By doing so, we aim to offer a more holistic understanding of the implications and nuances of e-contact in intergroup contexts.

With this response addressing the general comments, we can proceed to address each of the reviewer's specific comments in turn.

Main comments:

R2 Comment 2.

Based on how the introduction and theory sections are set up, I thought the CA was pretending to be a person from the other group, and the study was about having people chat with a bot who appeared to be a person from the outgroup. That is not the case. From the hypothesis, we later learn that the CA is merely facilitating the contact. So this study is less about testing the effects of e-contact, but more about comparing a CA-facilitated e-contact versus not facilitated. To bring this into Allport's conditions, it's like changing the condition between whether the contact has received support from some authority figure or not. The authors should be clear about this from the onset.

Response

We recognize the need for clarity in explaining the CA's function within our study, and we apologize for any confusion caused by the initial description. We have taken steps to enhance the transparency of this aspect. Specifically, we have added descriptions in introduction (lines 55-70) to clarify the CA's role as a facilitator of E-contact and its connection to Allport's conditions of authority support. This revision ensures that our study is accurately framed as a comparison between CA-facilitated and non-facilitated E-contact, akin to the condition of whether contact has received support from an authority figure or not, consistent with Allport's conditions.

Moreover, we have extended the "E-contact and Conversational Agent" section to provide a more comprehensive explanation of the CA's role as authority support and its implications within intergroup contact theory (lines 233-236). These revisions aim to address the reviewer's concerns and provide a clearer understanding of the study's focus.

R2 Comment 3.

Relatedly, the authors need to tackle this condition more in the theory. Even in the results and discussion sections, it's unclear why having a bot facilitate the discussion would lead to larger prejudice-reduction effects compared to not having that bot.

Response

We appreciate the reviewer's comment highlighting the need for a more in-depth exploration of the CA's role in our theoretical framework. To address this, we have expanded our theoretical section "E-Contact and Conversational Agents" and have described the theoretical foundation of the CA's role as a facilitator in intergroup E-contact (lines 223-270). We emphasize the CA's function as a form of authority support, which aligns with Allport's Contact Hypothesis. We have also addressed how we established such link through the consent process, recruitment, and platform instructions in our experiment. Additionally, we delved into how the CA's presence enhances social presence³, fosters human-like interaction and intensifies engagement⁴. This heightened engagement significantly influences attitudes and behaviors, creating a stronger sense of connection and familiarity among participants.

To substantiate these claims, we conducted statistical analyses confirming that participants in the treatment group, where a CA facilitated discussions, exhibited significantly higher social presence in terms of content contribution compared to those in the control group. These results are detailed

under a new subsection titled “Manipulation Check for the Impact of CA Facilitation on Social Presence” in the “Results” section (lines 616-626). Relatedly, in the “Discussion” section, we delved further into the mechanisms through which the CA might have reduced ethnic prejudice (lines 670-687). We highlighted the role of social presence and its connection to Allport's theory, specifically the concept of authority support.

These revisions aim to offer a clearer theoretical foundation for understanding the CA's potential impact on prejudice reduction.

R2 Comment 4.

Then again, how is the CA (particularly a bot) considered an authority figure? This is not what Allport had meant by institutional and authority support, right? Might the participants interpret this bot as connected to government or an NGO in some way? For future implementation, such as if an organization used this tool, the bot would have to appear as if it was coming from that organization, right?

Response

We appreciate the reviewer's thoughtful inquiry regarding the CA's role as an authority figure. We agree that Allport's original concept of authority support in the context of Contact Hypothesis focused on institutional and normative backing. In our study, we've applied a modern interpretation of this concept by utilizing a conversational agent (CA) to facilitate intergroup E-contact (Lines 74-77).

In the “E-Contact and Conversational Agents” section, we've expanded on the CA's role as a facilitator in intergroup E-contact (lines 223-236). While the CA in our study does not directly represent a government or an NGO, it is explicitly associated with the research institution conducting the study. This association is established through the consent process, recruitment procedures, and platform instructions. We acknowledge that participants may interpret the CA as an authority figure connected to the research institution.

Moreover, we believe that in the absence of an authority figure, unstructured conversations between antagonistic groups in an online platform can easily lead to digression, exacerbating fear, anxiety, and hostility (lines 58-67). The use of a chat moderator as an authority figure also has support in the literature (lines 228-231). For example, White, Turner et al.'s 2019¹ study employed contact conditions that connected ingroup and outgroup members through collaborative and goal-oriented online interactions, supervised by a chat moderator, as a form of authority support.

As for future implementations of this tool, especially when used by organizations, it would be important to tailor the design of the CA to align with the specific context and goals of the organization. This may involve incorporating textual cues that clearly establish a connection between the CA and the organization using the tool. By ensuring a clear representation of the CA's affiliation, organizations can maximize the potential of AI technologies in promoting positive intergroup relations within their respective contexts. The adaptability of the CA's role as an

authority figure is indeed an interesting area for further exploration, and your comment underscores the need for greater clarity in distinguishing the CA's role in our specific study context. We will consider this aspect in our future research endeavors to provide a more precise representation of authority support.

R2 Comment 5.

In the theory section, on pages 4-5, when authors write about voluntary participation and voluntary contact, eg. in the theory section on page 4, are they referring to self-selection, as in, if people self-select into contact with outgroups then they are more likely to feel less prejudiced? If that's the argument, it seems difficult to disentangle what the effects of contact are versus self-selection bias — these people are already less prejudiced and open to contact to begin with.

Response

We acknowledge the concern you have raised about potential difficulties in disentangling the effects of contact from self-selection bias, especially when individuals self-select into contact with outgroups. It's indeed a complex matter, and we want to clarify our perspective on this matter within the context of our study.

Our point of emphasis is that contact effects are stronger when an ingroup member willingly engages or interacts with members of other groups on equal terms (see^{15, p. 831}) compared to when they are forcefully brought together like the encounter between the colonizers and the colonized. The former holds better prospects for nurturing/developing friendship and trust compared to the latter (see^{9, p. 138}). Perhaps this is why a number of studies report that black-white intergroup relations tend to be characterized by greater anxiety and prospect of hostility due to experiences with slavery and colonization compared to other ethno-racial configurations that did not experience these historical processes (see¹⁶⁻¹⁹).

In practical terms, groups can have non-threatening contacts/interactions (with moderate level of anxiety (see^{13, p. 132}) like engaging in trade with one another or collaboratively working on a shared goal. This does not imply that the groups in question do not have some suspicion/anxiety about the motives of the other which is probably concealed. However, it becomes a different thing if the contact is threatening from the onset, as in when one group invades the territory or social boundary/space (through induced/forced contact) of the other. Which of these two kinds of contact hold greater prospect for positively transforming biases is obvious, particularly when viewed in light of Blumer's^{20, p. 4} four preconditions that make prejudice flourish. The literature also makes a distinction between spontaneous (voluntary) online contact and induced contact, demonstrating that the former holds greater positive transformational effects (see⁹).

R2 Comment 6.

On page 8, the authors write “E-contact also has the potential to generate unintended consequences,” but leave it at that. Can the authors go on to elaborate what those unintended (negative) consequences might be?

Response

Regarding the statement about the potential unintended consequences of E-contact on page 8 of our manuscript, we want to clarify that sentences addressing unintended consequences were removed from this section during a major revision based on reviewers' comments. However, we have elaborated on the unintended consequences of E-contact in the "Intergroup contact hypothesis: From theoretical propositions to empirical scrutiny" section, which can be found in lines (154-172) as detailed below.

E-contact could engender hostility, suspicion and increased reluctance to engage due to concern about anonymity, inadequate supervision and accountability as well as the possibility of limited restraint on the part of the interacting group members^{13,21}. Some scholars, particularly the Social Presence Theorists have also argued that the possibility of concealing non-verbal clues in online contact platforms hinders the prospect of developing more intimate interpersonal exchanges³. Commitments to online community could reinforce rather than assuage anti-outgroup biases where participants are under pressure to conform to the emerging normative expectations of their ingroups, particularly when such online communities are dominated by individuals from the same background^{9,10,22}. In such situations, online platforms become sites for circulating hate-speech and extremist predispositions that further fragment intergroup relationships²³. This is facilitated by people's tendency to associate with online communities with shared beliefs, values and other identity markers or symbolic representations^{13, p. 133}. Interaction through online platforms may compel participants to adopt a more politically correct interactional pattern that conceal their actual beliefs and sentiments in order to achieve actor's inter-relational needs¹.

A delay in responding to online conversation through disruption, technical glitch or the need to respond to emergency situations can heighten anxiety and tension, particularly when it involves individuals with different group affiliations^{13, p. 135, 24}.

R2 Comment 7.

It would be helpful to include examples of how the CA facilitated conversations. Much of the Discussion Instrument in the appendix could be in the main manuscript. And it would be helpful to have concrete examples of what these conversations and facilitators looked like.

Response

Thank you for your constructive feedback. We have taken your suggestions into account and have made the following revisions to our manuscript.

Firstly, we relocated the "Discussion Instrument" section, to the main body of the manuscript. Furthermore, we introduced an example (lines 447-459) that illustrates how the CA facilitated discussions. This example demonstrates the CA's role in categorizing participants' opinions, prompting discussions on issues, ideas, merits, and demerits, and guiding the conversation. We believe these changes significantly enhance the comprehensibility of our study and offer valuable insights into the CA's facilitation process in our research.

R2 Comment 8.

Using the CA implies deception — were participants aware that they were being facilitated by a computer program, not a real person? Who did they assume the CA was — Pashtun or non-Pashtun?

Response

Thank you for raising this important concern. We implemented several measures to prevent deception and ensure transparency in participant interactions. These measures aimed to provide participants with clear information about the presence and role of the CA, as well as assuring the confidentiality and anonymity of the discussions as described below.

Initial Communication for Transparency

To address the issue of deception from the outset, the recruiting agency directly contacted potential participants and explained how to use the discussion platform. They provided a comprehensive explanation of how to navigate the discussion platform, including details about group size. For groups with the CA, participants were also informed about its presence and role in guiding discussions. This initial communication set the stage for transparency and reduced the likelihood of participants perceiving the CA as a human participant.

Enhancing Transparency through Anonymization and Profiles

To enhance transparency and privacy, participants were organized into groups of four, and their identities were anonymized while indicating their respective ethnicities. Each participant was represented by a title (Mr./Ms.) followed by letters A to D along to indicate their ethnic groups, such as Mr./Ms. A/B/C/D Hazara/Pashtun/Tajik and was assigned a smiley face icon of a specific color. They were asked not to change their profile pictures or names during the discussion, which they complied with. The CA's profile had no titles or assigned letters. Its profile name was explicitly set as "AI facilitator". Importantly, the CA's profile picture prominently displayed the term "AI", clearly indicating its artificial intelligence nature. It's worth noting that we intentionally designed the CA to not represent any specific ethnic group, emphasizing its role as a neutral authority figure. Additionally, participants' education level, proficiency in English, and internet access were taken into account during the recruitment process. These factors contributed to their ability to recognize the CA as a non-human entity.

Clarifying CA's Role and Promoting Collaboration

In line with Allport's intergroup cooperation principle and to clarify the CA's role, we designed a task allocation system to promote collaboration. The theme description explicitly stated that each teamwork involved four participants, excluding the CA. In the three-day discussions, participants were assigned specific days for summarizing team opinions. For example, on day one, Mr. A was assigned the task of summarizing the team's opinions, and on subsequent days, other participants were similarly assigned.

The CA's role was distinct. While participants shared opinions, the CA focused on facilitating discussions using the Issue-Based Information System (IBIS) framework ²⁵ (please see "Facilitation instrument and setting" for more details on CA facilitation). This approach ensured that participants' opinions were guided without introducing the CA's views. This maintained

transparency and minimized confusion about the CA's role. Participants easily distinguished their own opinions from those guided by the AI, affirming that it was an automated system designed to enhance discussions. These measures prevented deception and promoted transparency, ensuring participants were aware of the CA's presence throughout the study

We have incorporated a dedicated subsection within the methods section that provides these details (lines 562-605).

R2 Comment 9.

Typically, this kind of deception brings many negative ethical implications, particularly in the context of the study if indeed prejudice against the outgroup was already so heightened. There could certainly be negative consequences if people realize that their facilitator was simply a computer program and not a real person, and any positive effects could vanish. How do the authors grapple with the ethical issues of the study?

Response

We appreciate your concern regarding the ethical implications. We agree that deception in research can raise ethical concerns. As described in the previous response, in our study, we were vigilant about addressing these concerns by emphasizing transparency from the outset. Participants were informed about the AI facilitator's presence and role. Our intention was to ensure that participants were aware that the AI facilitator was an automated system, and not a real person, to minimize the potential for deception.

However, as discussed in the newly added subsection "Ethical considerations in CA-facilitated intergroup contact" (lines 754-765) it is imperative for future research to transparently convey the nature of the facilitator to minimize the likelihood of participants mistakenly perceiving the CA facilitator as a human participant. The absence of such awareness could lead to unforeseen consequences, potentially eroding trust and cooperation²⁶.

R2 Comment 10.

The ethics section in the appendix right now just says the study was IRB approved. That is often just the bare minimum in ethical obligations for studying vulnerable populations.

Response

We appreciate your attention to the ethical aspects of our study. Ensuring the ethical conduct of research, particularly when working with vulnerable populations, is of utmost importance to us. In our study, we implemented a rigorous ethical framework that included multiple layers of safeguarding the participants' rights and well-being.

Informed Consent: Participants provided informed consent at multiple points in the study, including before and after the discussions. The consent forms explicitly outlined the study's procedures, its purpose, and the measures in place to protect participants' confidentiality and privacy. While the specific research hypothesis was not disclosed to prevent potential bias,

participants were made aware of the study's broader goals related to promoting interethnic understanding.

Privacy and Anonymity: To protect participants' privacy and anonymity, we assigned them pseudonyms (Mr./Ms. A/B/C/D) and provided clear instructions to not change their profile pictures or names. Discussion topics were designed to avoid the need for sharing personal information, further safeguarding participants' identities.

Transparency: We maintained transparency throughout the study by clearly informing participants about the AI facilitator's presence and role in guiding discussions as stated in previous responses. This was reinforced through the profiles and names assigned to both participants and the AI facilitator.

Debriefing: While participants were informed about the overall goal of promoting positive intergroup interactions among different ethnic groups, we refrained from sharing the specific research hypothesis. This precaution was taken due to the sensitive context in Afghanistan, where any misinterpretation of our intentions could have posed risks to participants and the recruiting agency.

We have expanded the ethics section (lines 562-605), to offer detailed account of the measures we implemented to safeguard participants and uphold ethical standards.

R2 Comment 11.

As an implication for this study, what if the CA appears to show bias? We know that AI often reproduces existing societal biases. Could this be a realistic concern for this study, and future possible implementation of this tool?

Response

Acknowledging the potential for AI systems to perpetuate societal biases is a critical consideration²⁷. This realistic concern can indeed impact study outcomes and the future use of this tool. To address this concern, we emphasize the importance of continuous monitoring and evaluation of the CA²⁸. This ongoing assessment should scrutinize the CA's responses, prompts, and interactions, with the goal of detecting and mitigating the risk of reinforcing prejudiced attitudes or behaviors.

In our study, we took several meticulous measures to minimize bias in the CA's facilitation settings:

1. **Content-Driven Prompts:** The CA's messages in D-Agree are generated based on the content of participants' opinions rather than their ethnic backgrounds. The facilitation messages align with the Issue-based Information System (IBIS) framework, categorizing participants' opinions as ideas, issues, pros, or cons and providing relevant facilitation messages accordingly (See Methods). This approach aims to foster balanced and inclusive discussions.

2. **Equal Distribution of Involvement:** The CA's involvement is distributed evenly across discussions to prevent any potential concentration of facilitation toward a particular ethnic group. Specifically, for every three opinions expressed, the CA provides facilitation for one opinion, regardless of who wrote the opinions.

However, it's important to note that while the CA categorizes opinions, it doesn't directly recognize biased content within discussions. To comprehensively address this concern in future studies, we suggest:

- (a) **Enhancing CA Capabilities:** Explore the possibility of enhancing the CA's ability to recognize and address biased content, especially in scenarios where participants express prejudiced viewpoints. This might involve refining algorithms to identify biased language or ideas, while also ensuring that the CA's responses remain unbiased and impartial.
- (b) **Implementing Reporting Mechanisms:** Implement mechanisms within the discussion platform for participants to report instances of perceived bias or biased content. This feedback can be valuable in identifying and rectifying bias-related issues promptly.

We have added such descriptions in the “Ethical considerations in CA-facilitated intergroup contact” (lines 766-780). These measures collectively aim to ensure fair and unbiased facilitation in discussions facilitated by the CA. Acknowledging that AI systems can perpetuate societal biases is crucial, and the outlined steps of regular monitoring, evaluation, and active mitigation strategies are integral to addressing this concern, both in our study and in future applications.

R2 Comment 12.

Was it not a possibility to actually allow for e-contact between participants from both groups, and include a trained enumerator to facilitate? Why use a bot? Or why not compare a human versus bot facilitator?

Response

While we acknowledge that previous research has utilized pre-programmed (human) chat moderators in e-contact experiments^{1,2}, and we also acknowledge the potential for human facilitators in future studies or comparing AI with human facilitators, we chose to use AI in this study for several compelling reasons, as outlined below and also briefly incorporated in the “Ethical considerations in CA-facilitated intergroup contact” (lines 745-753) subsection.

Bias mitigation: Using an AI as a facilitator significantly reduces the potential for bias or perceived bias²⁹, especially in contexts where ethnic tensions are high. Unlike humans, AI facilitators are not influenced by personal biases or external factors that could impact their interactions with different ethnic groups.

Privacy and confidentiality: AI facilitators excel in maintaining participant privacy and confidentiality. Since AI does not possess personal biases or motivations, participants may feel

more comfortable sharing their thoughts and experiences without concerns about potential judgment or disclosure. AI systems can be designed to adhere to data privacy regulations and ensure the confidentiality of participants' information.

Consistency: AI facilitators consistently apply predetermined rules and guidelines across all interactions. They can provide standardized responses and ensure equal treatment for all participants. This consistency helps to create a fair and level playing field for engaging in intergroup contact.

Social desirability and psychological safety: AI facilitators may mitigate social desirability bias or the Hawthorne effect³⁰, as participants perceive AI as non-judgmental and unbiased. This fosters a more relaxed and less pressured environment, allowing participants to express their true thoughts and opinions without conforming to societal expectations. The consistent and standardized nature of AI facilitators' responses further contributes to a sense of psychological safety.

Scalability: AI facilitators have the distinct advantage of handling a large number of interactions simultaneously, making them suitable for facilitating contacts on a large scale. They can engage with multiple participants simultaneously, which is especially valuable in scenarios where a large number of individuals are involved or where contact needs to occur across various platforms or channels.

However, we fully acknowledge the need for further research to compare the effectiveness of AI facilitators versus human facilitators, as well as explore other potential settings and approaches. This consideration for future research will help shed more light on the comparative advantages and limitations of different facilitation methods in reducing prejudice.

R2 Comment 13.

What about concerns around social desirability? With the facilitator, participants might have felt more pressure to answer in socially desirable ways since they knew they were being observed (Hawthorne effect).

Response

We acknowledge the potential influence of social desirability bias in participants' responses, particularly due to their awareness of being observed during the e-contact sessions (known as the Hawthorne effect). While this concern is relevant, our research design aimed to address it by employing AI facilitators instead of human facilitators. AI facilitators can create a more relaxed and less pressured environment for participants, potentially reducing the impact of social desirability bias. Participants may perceive AI as non-judgmental and unbiased, enabling them to express their genuine thoughts and opinions.

However, it is important to acknowledge that further research is necessary to comprehensively explore the effects of social desirability and the Hawthorne effect in the context of e-contact experiments facilitated by AI. This research could involve direct comparisons between AI and human facilitators, as well as investigating the specific mechanisms through which AI facilitators

contribute to reducing bias and creating a conducive environment for intergroup contact. In response to this comment, we have also addressed the issue of social desirability in our “Ethical considerations in CA-facilitated intergroup contact” section (lines 781-786).

R2 Comment 14.

In terms of work that should be cited: see Nejla Asimovic’s research around social media facilitated intergroup contact, Salma Mousa’s e-contact research as well as her 2020 Science paper that shows contact with outgroup members may only change attitudes about those specific members not the outgroup as a whole. Lastly, Zhou and Lyall 2023 used an RCT to test a large scale, prolonged contact intervention in Kandahar, Afghanistan and found null effects.

Response

We appreciate your suggestions regarding additional citations. Here are the actions we have taken:

Citation of Nejla Asimovic's Research: We have included a citation to Asimovic et al. (2021)²³ in the manuscript, which addresses the role of online platforms in exacerbating intergroup tensions and extremist tendencies. This addition can be found in lines 163-165.

Citation of Salma Mousa's Work: It's worth noting that we had previously cited Mousa's work²² in our manuscript. Additionally, we have now added a citation to Mousa's 2020 Science paper³¹, which aligns with the limitations of intergroup contact discussed in our paper. You can find this citation in lines 115-118.

Citation of Zhou and Lyall (2023): We have also included citations to Zhou and Lyall (2022)⁷ in our manuscript to address one of the two studies conducted in Afghanistan regarding intergroup relations and the contact hypothesis, which resulted in null results (lines 663-664). Other citations can be found in lines 55, 661, 709 and 724.

We sincerely appreciate your input, which has contributed to the enhancement of our manuscript's quality.

R2 Comment 15.

What are the scope conditions of this study? While the authors argue that Afghanistan is a lesser studied context for intergroup contact research, what should we learn from this context specifically? How does this new context add other theoretical or empirical contributions? Do the authors anticipate that their intervention would work more easily in a western context or in a place that was not experiencing active conflict? They should also discuss in greater detail how ongoing violence and political conflict might shape how the intervention is designed and its success?

Response

The scope conditions of this study encompass the unique context of Afghanistan, a relatively understudied setting for intergroup contact research. Afghanistan's specific socio-political landscape, characterized by ongoing violence and conflict, adds distinctiveness to this study.

Examining the effectiveness of indirect contact mechanisms for reducing prejudice in such a complex and conflict-ridden environment is crucial for several reasons.

Firstly, Afghanistan provides a valuable test case for the generalizability of intergroup contact theories beyond Western relatively stable sociopolitical context. This study aims to contribute to the growing body of literature on intergroup contact by demonstrating the adaptability and applicability of contact-based interventions in non-Western, conflict-affected settings. The findings from Afghanistan can potentially inform similar interventions in regions facing similar challenges.

Moreover, the study aims to uncover how the Afghan context adds theoretical and empirical contributions to the field of intergroup contact. By exploring intergroup contact in a context marked by active conflict, we can gain insights into the mechanisms that drive contact's effectiveness in reducing prejudice, even in the face of significant adversity. This understanding can enrich existing theories of intergroup contact and help refine interventions for conflict-affected regions worldwide.

While the primary focus of this study was to evaluate the intervention's effectiveness in the Afghan context, it's essential to clarify that the study did not make specific predictions regarding the intervention's applicability in a Western context or in places not experiencing active conflict. Instead, the study aimed to provide insights into the effects of the intervention in a conflict-afflicted setting. However, the findings have broader implications that extend beyond conflict-affected areas. They can enhance our understanding of intergroup dynamics and prejudice reduction efforts in diverse contexts characterized by tensions, which may encompass both conflict-affected regions and Western societies during periods of heightened tension. For instance, the study's insights may be particularly relevant in addressing prejudice towards various groups. This includes situations such as anti-Muslim sentiments following terrorist attacks by extremist groups like ISIL, increased bias targeted at Asians during the COVID-19 pandemic, or the prejudice that accompanied the 2022 invasion of Ukraine by Russia. Therefore, while this study was conducted in a specific context, its findings offer valuable insights that are applicable to a range of scenarios marked by intergroup tensions. We have introduced a new subsection in the discussion section titled "Scope and Generalizability of Findings" to address these points (lines 706-732).

The ongoing conflict in Afghanistan significantly influenced our intervention and research approach. Afghanistan's unique 'in-conflict' situation, characterized by heightened interethnic tensions following the forceful seizure of power by the Taliban and the dominance of Pashtuns in its sociopolitical affairs, presented significant challenges. Conducting direct face-to-face contact was logistically challenging and, in terms of safety, impractical, given the volatile situation. However, even more crucial was the risk of exacerbating existing biases and tensions⁶, especially without proper supervision. To address these challenges, we conducted E-contact and introduced a Conversational Agent (CA) as an impartial mediator of intergroup contact, free from ethnic affiliations. This CA-facilitated E-contact allowed us to transcend geographical barriers and provided a platform for individuals from diverse ethnic backgrounds to engage in dialogue, share perspectives, and challenge previously held stereotypes. This approach proved to be a practical and safe alternative in the context of Afghanistan's complex sociopolitical environment.

Our research design prioritized minimizing potential harm, ensuring participant confidentiality and anonymity, and fostering an environment conducive to open and constructive intergroup dialogue. Despite acknowledging the limitations and challenges of conducting research in a conflict-affected setting, our commitment to advancing our understanding of intergroup dynamics using technology-driven interventions remains steadfast.

In summary, while the scope conditions of this study focused on Afghanistan, they contribute to a broader understanding of intergroup contact's applicability and effectiveness in other non-Western, conflict-affected contexts. The discussion of how ongoing conflict shapes the intervention's design and success is part of the study's research objectives, and the findings will provide insights into these dynamics. We have addressed this in the "Intergroup Relations in Afghanistan" section (see lines 301-308) for context.

Minor comments:

R2 Comment 16.

In the abstract and intro, describe the magnitude of the effects. Hard to gauge with simply positive and statistically significant.

Response

We have revised the abstract (lines 14-15) and introduction (line 79) to describe the modest magnitude of the effects more explicitly.

R2 Comment 17.

These sentences were confusing:

On p4: "In other words, the more intimate contacts that members of an ingroup have with members of an outgroup, the more positive their attitudes and relationship with such outgroup and indeed other outgroups not necessarily encountered in the initial contact situation."

Are the authors trying to say that positive contact with a few members of an outgroup can less prejudice against that outgroup as a whole? Or that Allport's conditions don't necessarily need to be met?

On p5: "However, despite the widespread appeal that the contact hypothesis has received in academic and policy circles, some scholars have cautioned against the credulous association of intergroup contact with positive outcomes."

Response

Thank you for your feedback on the sentences in question. We understand your concerns about potential confusion. As part of our manuscript's major revision, we deleted the first confusing sentences (p4) mentioned. However, we would like to provide clarifications here.

This is a matter of presenting the divergent concerns of different scholars in the field of intergroup contacts/relations-some optimistic, and others pessimistic about the efficacy of intergroup contact

in different contexts. Our position is also shared by some scholars like Walther, Hoyer, Ganayem and Shonfeld (2015)^{32, p. 556}. For example, Mousa (2020)³¹ reported that although intergroup contacts improved relationships between ingroups and outgroups engaged in the intervention, the transferability of these improved measures of intergroup relationship beyond the intervention remains doubtful and cumbersome. Similarly, Dixon, (2006)¹⁷ argues that while mere contacts between particular groups have been reported to ease intergroup tensions and reduce prejudicial attitudes, more needs to be done to assuage the anxiety felt by groups with a historically and institutionally embedded antipathy and hostility like whites and blacks.

Thus, different intergroup relationships may elicit varying degrees of threats or contact effects when we factor in the characteristics of individual members in intergroup relationships, the groups ideological positioning, economic, sociocultural and institutional peculiarities^{16,19,33}. These factors explain why it is difficult to assume a straightforward positive connection between intergroup contact and prejudice reduction/improved understanding/intergroup harmony. Indeed, contexts with a great deal of ethno-racial diversity often report alarming levels of intergroup intolerance and hostility despite greater possibility of contacts^{34,35}, suggesting a multi-pronged approach to prejudice reduction since the effects of intergroup contacts are not unidirectional

In addition, the latter sentence (p.5) is supported by debates in the literature on intergroup contact generally. Recall that two main theoretical perspectives: Group Threat Theory (GTT) and Intergroup Contact Hypothesis (IGCH) are often used to make sense of the processes that unfold when two or more groups come into contact³⁶. The GTT, which is associated with scholars like Blumer (1958)²⁰ and Blalock (1967)³⁷ contends that competing and antagonistic interests among individuals and groups is the major driving force of intergroup relationships³⁸⁻⁴¹. Accordingly, actual or perceived competition over scarce economic, political, cultural or social resources between minority and majority groups generates a considerable level of insecurity and threat within the latter group to the extent that prejudice, resentment and discrimination become adaptive responses^{20,37,42}. Blumer (1958)^{20, p. 6} opines that ethno-racial hierarchy and its accompanying stereotypes targeted at minority groups are not merely the outcomes of direct contacts, but deeply embedded in the majority group's history and cultural heritage. Intergroup threat also prevails in the expression of cultural values with different groups seeking to defend its cherished racial and cultural attributes from encroachment or dilution from outgroup members⁴³⁻⁴⁵.

In conclusion, the Group Threat Theory essentially believes that to the extent that contacts often increase intermingling and competition between groups, threat persists due to divided interests about allocation of and access to scarce resources. This makes it difficult to assume a straightforward link between contact and sustained intergroup harmony.

Reviewer 3

Reducing Interethnic Bias with Conversational Agent in Contact Experiments in Afghanistan

The authors point out that while much work has been done over the years to examine the contact hypothesis through direct forms of inter-group exposure, less has been done to examine more indirect contact, especially involving new online technologies. Using a multi-stage indirect contact experiment among Afghans, the authors find that conversation agents (CA) on online discussion platforms reduce interethnic prejudice. I believe the manuscript could potentially be published in *Communications Psychology* but it would require major revisions, but given the extent of those revisions, I recommend that the manuscript either be rejected or “rejected and resubmitted”.

Response

We appreciate your review and the recognition of the novelty of our study. We acknowledge your recommendation for major revisions and assure you that we are committed to addressing all your concerns to enhance the manuscript's quality. Your feedback is invaluable, and we have worked diligently to improve the clarity, organization, and statistical robustness of our research.

R3 Comment 1.

Clarify the manuscript's contribution. The authors state in the literature review that “A meta-analysis conducted by Imperato and others (2021) on the effects of virtual platforms on prejudice reduction revealed that like direct contact, contact through online platforms had significant positive effects on the transformation of prejudicial attitudes notwithstanding the different samples involved and the research context.” Identify this manuscript's contribution in light of existing literature on the topic more clearly. The discussion in lines 232-241 on the manuscript's contribution should come earlier.

Response

Thank you for your feedback. In response to the suggestion to clarify the manuscript's contribution, we have made specific additions and revisions in the “Introduction” section of the manuscript (lines 188-206). We also moved the discussion of our manuscript's contributions to “Intergroup contact hypothesis: From theoretical propositions to empirical scrutiny” section (lines 198-206). Below, we provide some additional clarifications.

Our contribution extends the literature on intergroup contact effects by highlighting a particular condition (using a CA vs not using a CA to facilitate E-contact/interaction) under which contact may yield more fruitful outcomes, thereby directly contributing to broadening our understanding of how intergroup contact and expected outcomes emanate^{1,13,46}. For example, Condra and Linardi (2019)⁶ study in Afghanistan showed that unstructured and unsupervised intergroup contacts/interactions may escalate rather than assuage intergroup anxiety and stereotypes, suggesting that researchers should pay greater attention to the conditions surrounding the enactment of intergroup contacts/interaction. Therefore, we argue that having a CA facilitate

online discussions is a potentially cost-efficient and relatively unbiased strategy for a more fruitful intergroup contact outcome in online platforms.

A major issue with the use of online contact platforms is deficient supervision^{1,6,13,46}. The use of a conversational agent in online discussion platform therefore holds prospect for bridging this gap by guiding and moderating the conversation to achieved desired goals.

R3 Comment 2.

Direct vs. Indirect contact discussion. Some of the manuscript's focus on direct vs. indirect contact effects seems tangential given that the authors are not evaluating direct contact effects.

Response

Reponses are provided collectively under R3 Comment 3

R3 Comment 3.

Theorizing positive effects of online indirect contact. Instead, the authors should focus more on the controversy surrounding indirect contact online, and whether it reduces or enhances prejudice. The authors should provide a theoretical framework to explain when indirect contact should work to reduce prejudice online and test hypotheses using the case at hand. Identify key scope conditions in relation to CAs. What do CAs represent? They signal "rules of the game" for online discussion, but they do not have monitoring or enforcement power over group behavior, so they are weak institutions at best. They are also exogenous, and it's unclear whether groups would adopt CAs on their own in nature. Still, if weak institutions can reduce prejudice online in relatively non-costly ways, then the study has useful, practical policy implications for internet policing.

Response

In light of the varying perspectives on the discussion of direct and indirect intergroup contact, we appreciate your feedback. Reviewer 1 emphasized the need for a clear distinction between direct and indirect contact, while Reviewer 3 suggested focusing more on the controversy surrounding indirect contact's impact on prejudice reduction in online contexts. To address these concerns effectively, we've made specific revisions in the manuscript. Lines 131-172 address various perspectives on E-contact, including both positive and negative aspects. We delve into the challenges and potential negative outcomes of indirect contact in online contexts (lines 154-172), contributing to the ongoing debate about its effectiveness in prejudice reduction. We discuss the concerns related to anonymity, limited supervision, accountability, and the challenges of concealing non-verbal cues in online platforms^{13,21}. Furthermore, we acknowledge the potential fragmentation of intergroup relationships within online communities dominated by individuals with similar backgrounds and beliefs¹³. This discussion emphasizes the capacity of online platforms to facilitate hate speech and extremist predispositions²³. Additionally, we consider how participants in online interactions may adopt politically correct interaction patterns that conceal their true beliefs¹ and the potential for delays in responding to online conversations^{13,24}.

Regarding the role of Conversational Agents (CAs), we provide extensive explanations in the "Introduction" (lines 62-77) "Intergroup contact hypothesis: From theoretical propositions to

empirical scrutiny" (lines 188-195) and specifically in "E-Contact and Conversational Agents" sections (lines 223-265). In our study, CAs assume the role of authority support, as advocated by Allport's theory, by linking it to the research institution through consent processes, recruitment, and platform instructions. This positioning aligns with the concept of authority support and suggests that influential figures or institutions can foster positive intergroup interactions.

We also address the key scope conditions of CAs, acknowledging their limitations as weak institutions in the "Ethical considerations in CA-facilitated intergroup contact" sub-section (lines 734-741). Additionally, we recognize the exogenous nature of CAs and the challenge of their adoption in real-world scenarios, which is highlighted as a limitation in the "Research Limitations" section (lines 811-813).

Furthermore, we discuss the practical implications of our study for online discussions and potential internet policing strategies in the "Research findings and implications" subsection (lines 694-699). This approach offers a cost-effective means to reduce online prejudice, particularly in situations where direct enforcement may be challenging, making it timely and relevant in various post-conflict scenarios.

These revisions and clarifications in the manuscript aim to address the concerns raised by Reviewer 3 and provide a more balanced and informative discussion.

R3 Comment 4.

Rationale for Afghanistan. Why study this case? What makes Afghanistan compelling for examining indirect contact mechanisms for reducing prejudice? This needs clarification. How generalizable are the results to other contexts?

Response

The choice to study Afghanistan as a context for examining indirect contact mechanisms for reducing prejudice is underpinned by several compelling reasons. Firstly, Afghanistan represents a unique and less-explored setting for intergroup contact research due to its ongoing conflict and ethnic diversity. This context provides an opportunity to extend the understanding of intergroup dynamics and the effectiveness of contact-based interventions to non-Western, conflict-affected regions. While intergroup contact research has primarily focused on Western, relatively stable contexts, Afghanistan's complex socio-political landscape allows us to explore how these mechanisms operate under challenging conditions.

Secondly, Afghanistan's unique 'in-conflict' situation, characterized by heightened interethnic tensions emerged following the forceful seizure of power by the Taliban in 2021 and the dominance of Pashtuns in its sociopolitical affairs. Conducting direct face-to-face contact was logistically challenging and, in terms of safety, impractical, given the volatile situation. However, even more crucial was the risk of exacerbating existing biases and tensions⁶, especially without proper supervision. To address these challenges, we conducted E-contact and introduced a Conversational Agent (CA) as an impartial mediator of intergroup contact, free from ethnic affiliations. This CA-facilitated E-contact allowed us to transcend geographical barriers and

provided a platform for individuals from diverse ethnic backgrounds to engage in dialogue, share perspectives, and challenge previously held stereotypes. This approach proved to be a practical and safe alternative in the context of Afghanistan's complex sociopolitical environment. We have addressed these in the "Intergroup Relations in Afghanistan" section (see lines 301-304).

As for the generalizability of the results, while this study is rooted in the Afghan context, its findings hold valuable insights for a range of other contexts characterized by intergroup tensions. The study's focus on a conflict-affected setting allows us to generate theoretical and empirical contributions applicable to similar regions facing similar challenges. The lessons learned from Afghanistan can inform interventions in areas grappling with intergroup prejudices stemming from persistent conflicts, historical divisions, political polarization, and proliferation of extremist ideologies. We have added a new subsection in the Discussion section "Scope and Generalizability of Findings" and have discussed these in detail (lines 706-732).

In summary, Afghanistan's unique combination of conflict, ethnic diversity, and power dynamics makes it a compelling case for studying indirect contact mechanisms for reducing prejudice. While the study's primary focus is on Afghanistan, its broader implications extend to understanding intergroup dynamics and prejudice reduction efforts in diverse contexts where tensions exist, thus enhancing its generalizability beyond the Afghan context.

R3 Comment 5.

Hypotheses: H1 and H2 seem the same to me. H3 is the null. Why not just simplify this to one hypothesis on the effectiveness of CAs in reducing prejudice? H2 and H3 are not needed. H4 is not well-theorized and needs further development to understand the causal process of mediation. I think what the authors really argue is that anxiety is a moderator, not a causal mediator.

Response

Thank you for your feedback on our hypotheses. We appreciate your input regarding the simplification of our hypotheses. In response to your suggestion, we have simplified our hypotheses into one comprehensive statement: "H1: Intergroup/ethnic discussions in online platforms, which are facilitated by CAs are more likely to reduce mutual prejudice than unfacilitated online discussions."

We agree that the previous set of hypotheses might have introduced unnecessary complexity. By streamlining our hypotheses, we aim to provide a more focused and coherent framework for our study.

R3 Comment 6.

Research Design. The authors need a clear research design section that establishes the experimental design, how it tests the hypotheses and identifies the causal effects of CA. This is currently missing. This makes the Results section all the more difficult to follow. The presentation is more like a mystery novel, where we are focused to infer the design from the discussion of results. Time effects in the design are also not theorized or hypothesized clearly. What is the CA? How is it operationalized? So much is unclear.

Response

We appreciate your feedback regarding the need for a clear research design section in our manuscript. We deeply regret any confusion or lack of clarity in our initial submission.

To address your concerns, we have implemented significant improvements to the manuscript. We have introduced a comprehensive methods section in the main manuscript, which provides a detailed account of our research design, measures, procedures, and other relevant information.

In response to your questions about the CA, we have included a dedicated subsection titled "Study instrument" within the methods section, where we explain how the CA operates. Additionally, we have elaborated on the concept of CA and its significance in the "E-Contact and Conversational Agents" section, providing readers with a comprehensive understanding of CA and its role within our research.

Regarding time effects in the design, we have addressed this aspect in the "The current study" section (lines 322-325).

These revisions aim to enhance the experimental design, hypothesis testing methodology, and CA's causal effect identification for improved clarity and readability. Your guidance is greatly appreciated.

R3 Comment 7.

Results. The presentation of results should focus on testing key hypotheses about CA effects. How are intergroup prejudices measured? Table 1 suggests to me that CAs have a potentially weak reductive effect. Figure 1 seems unnecessary. Why is this even in the manuscript? It doesn't establish a clear mechanism in relation to CAs on reducing prejudice (or anxiety). The mechanism is unclear. Also, it's unclear how either of these things are measured empirically.

Response

We appreciate your valuable feedback concerning the presentation of our research results and the need to focus on the core hypotheses related to the effects of Conversational Agents (CAs). In response to your insightful comments, we took several steps to enhance the clarity and relevance of our manuscript. First, we consolidated our multiple hypotheses into a single overarching hypothesis, as previously mentioned. Additionally, we streamlined our manuscript by removing the mediation analysis of intergroup anxiety on prejudice reduction, including Figure 1, due to null results. These changes allow us to maintain a sharper focus on our primary hypotheses and key findings.

Regarding the mechanism through which CAs reduce prejudice, our study incorporates the concept of authority support, which aligns with Allport's Contact Hypothesis theory. Authority support describes the role of institutional or organizational endorsement in fostering positive intergroup interactions⁸. In our study, the presence of a CA can be viewed as a form of authority

support, which facilitates higher levels of social presence within the discussion groups. We assessed social presence by measuring the number of ideas and the length of opinions contributed by participants in the online discussion, and our manipulation check further confirmed that CA-facilitated discussions exhibited significantly higher social presence compared to control groups. This heightened social presence, driven by the CA's role, is integral in reducing intergroup anxiety and prejudice.

We have addressed these aspects in “E-Contact and Conversational Agents” alongside other relevant considerations. Additionally, we have incorporated a “Manipulation Check for the Impact of CA Facilitation on Social Presence” in the results section and provided a detailed discussion of these findings in the “Discussion” section.

While we acknowledge that the CA had a modest effect, it gains particular significance when considered within the unique context of Afghanistan, characterized by ongoing conflict. This context has been previously associated with heightened prejudice⁶ or inconclusive findings⁷ in intergroup research. Furthermore, these effect sizes compare with those reported in broader studies, including Pettigrew's influential 2006⁸ meta-analysis (mean r s range of 0.205 to 0.214, characterized as small to medium medium) and Imperato, Schneider et al.'s⁹ comprehensive 2021 meta-analysis of 23 E-contact studies ($d = 0.36$, characterized as medium). We have addressed these in the Discussion Section (lines 662-669).

We have provided details on how we measured intergroup anxiety and intergroup prejudice in the "Measures" subsection (lines 382-401) of Methods. We employed the Bogardus (1925)⁴⁷ social distance scale to measure intergroup prejudice among participants. Intergroup anxiety was measured using an adapted version of Stephan and Stephan's (1985)⁴⁸ Intergroup Anxiety Scale, tailored for non-native English speakers⁴⁹.

It is important to reiterate that our study was conducted in Afghanistan during a particularly challenging period when the Taliban were in control. Given this sensitive context and our concerns for participants' safety, we meticulously designed our research questions, measures, and procedures. These constraints shaped our approach and are acknowledged as part of our study's context.

R3 Comment 8.

Discussion. Given the preceding problems, it is difficult to draw implications from the study in the discussion that follows. The section at the end on study limitations raises questions about sample size (not reported in the manuscript), sampling bias (who are the participants?), the consent process (was deception involved? Were respondents debriefed? Did the study have IRB approval? When was it conducted? Ethical implications?). This should come before the discussion section, ideally as part of a section on data collection that precedes the results.

Response

We acknowledge that our manuscript required significant revision to meet the standards expected for publication. We conducted extensive revisions to address the reviewer's comments, which substantially improved our manuscript.

In response to the concerns raised about sample size, sampling bias, the consent process, ethical considerations, deception, and the timing of our study, we have incorporated a comprehensive methods section into the main manuscript. In this section, we provided detailed explanations and information pertaining to these aspects to address the reviewer's queries.

Furthermore, we expanded our discussion to include a new section titled "Ethical Considerations in CA-Facilitated Intergroup Contact." This section delved into the implications of employing a CA in intergroup contact and provided a thorough exploration of the ethical implications and considerations.

Additionally, we added the "Ethical Considerations" section in methods, to include detailed information on compliance with informed consent, privacy, anonymization, transparency, and debriefing. This section also covered the study's IRB approval information.

By addressing these issues in both the methods and discussion sections, we enhanced the transparency and completeness of our manuscript. We also expanded our discussion of the research findings and their practical implications to provide a more comprehensive understanding of the study's significance.

R3 Comment 9.

In summary, my initial enthusiasm to review this manuscript based on the abstract has been dampened by disappointment in the execution of the manuscript for the reasons stated above. I could go into other issues, but I trust that other reviewers will raise similar points. My comments should not discourage the authors from revising the manuscript, but the authors should really work on organizing the manuscript before resubmitting it either here or elsewhere. Finally, here are questions that the authors should especially focus on from the editors of CP.

Response

We appreciate your time and valuable feedback on our manuscript. We understand your concerns regarding the execution and organization of the manuscript. We apologize for any inconveniences caused and assure you that we have carefully considered your comments and those of the other reviewers. Based on your feedback, we made significant improvements in the revised version of the manuscript.

Regarding the questions posed by the editors of CP:

R3 Comment 10.

-Does the paper represent an advance in understanding which may influence thinking in the field?
I would argue yes potentially.

Response

We appreciate the reviewers' recognition of the potential impact of our study in advancing understanding knowledge in the field. Our research explored the innovative use of conversational agents (CAs) as a tool to reduce intergroup prejudice and anxiety, especially in challenging contexts like Afghanistan. We recognize the reviewers' optimism about the potential influence of our work in the field of intergroup relations and communication.

In response to the reviewer's feedback, we made significant revisions to the manuscript to address concerns and enhance the clarity and rigor of our study. By providing more comprehensive explanations of our methodology, measures, and ethical considerations, we aimed to strengthen the potential impact of our research and its contribution to the field's thinking and practices.

R3 Comment 11.

-Does the article presents an original study, new analysis, new model, or a direct or extended replication of previous work? An original study, clearly.

Response

We appreciate the reviewer's acknowledgment that our study represents an original contribution to the field. In order to enhance the clarity and rigor of our study, we revised the manuscript to provide a more comprehensive and transparent presentation of our research, including early description of our research novelty and contribution in light of existing literature.

R3 Comment 12.

-Are the data and analysis technically sound? Are they appropriate to answer the research question, e.g., are causal research questions addressed on the basis of causal, rather than correlational evidence? Here there is real room for improvement.

We acknowledge the importance of addressing causal research questions in our study. While we initially did not clearly describe our research design in the manuscript, we have now made significant improvements. In the research design section of the methods (lines 368-370), we have explicitly detailed our use of a randomized controlled trial (RCT) design. This RCT approach is well-suited for establishing causal relationships as participants were randomly assigned to two conditions: CA facilitation versus no CA facilitation. These changes enhance the clarity of our research design and aim to provide more robust evidence of causal effects

R3 Comment 13.

-Does the paper provide strong evidence for its conclusions? The causal effects seem weak at best.

Response

We acknowledge that causal inference in social sciences, especially in the context of intergroup dynamics and complex human interactions, can be challenging. While our study aimed to

establish causal relationships between CA facilitation and the reduction of intergroup prejudice and anxiety, we recognize that the causal effects observed may appear modest.

It's crucial to consider the broader context of our study, which was conducted in Afghanistan during a period marked by heightened intergroup tensions and the re-emergence of the Taliban. These contextual factors posed unique challenges to the research design and may have influenced the observed effects. Given these challenges, we believe that our study contributes valuable insights into the potential of using CAs in intergroup contacts, particularly in volatile settings.

We revised our manuscript to emphasize the significance of our findings within this challenging context and to acknowledge the limitations of our study. In the revised discussion section, we provided a more nuanced interpretation of the observed effects and their implications, emphasizing the potential for CAs to serve as a valuable tool for promoting positive intergroup interactions, even if the effects was not overwhelmingly strong.

In summary, while we acknowledge that the causal effects observed in our study may appear modest, we believe that the evidence presented contributes to our understanding of the role of CAs in intergroup contact, especially in challenging and sensitive environments like Afghanistan. Our revised manuscript provides a more balanced and context-aware discussion of the strength of the evidence and its implications.

R3 Comment 14.

-Is the study question important to scientists for a sub-field of psychology? Potentially

Response

We appreciate the reviewer's consideration of the importance of our study's question in the field of intergroup relations. Our research addresses the role of conversational agents (CAs) in facilitating intergroup contact, a timely and relevant topic in the context of technology-mediated interactions. Our revised manuscript discusses the significance of our research question and its implications for the field of intergroup relations.

R3 Comment 15.

-Are there any special ethical concerns arising from the use of animals or human subjects? Potentially, these are not addressed clearly in the manuscript.

Response

We acknowledge the reviewer's valid concern about the clarity of ethical considerations in our manuscript. Ethical considerations are of utmost importance in research involving human subjects, and we appreciate the opportunity to provide a more detailed account of how we addressed them.

Our study was conducted in a sensitive and challenging context, Afghanistan, where the safety and well-being of participants were paramount. To address potential ethical concerns, we implemented several measures:

Informed Consent: Informed consent was obtained from all participants at multiple key points during the study. Participants received comprehensive information about the study's procedures, objectives, and expected outcomes prior to the research. This included details about the various study components, such as surveys and online discussions.

Privacy and Anonymity: To ensure privacy and anonymity, participants' identities were anonymized while indicating their respective ethnicities. Each participant was represented by a title (Mr./Ms.) followed by letters A to D and was assigned a smiley face icon of a specific color.

Transparency and Debriefing: Participants were informed about the study's overarching goal of promoting positive intergroup interactions among different ethnic groups during the debriefing phase. However, we refrained from disclosing the specific research hypothesis due to the sensitive context in Afghanistan, where any misinterpretation of our intentions could have posed risks to participants and the recruiting agency.

No Deception: We maintained transparency and minimized the potential for deception by informing participants from the outset that they would interact with an AI facilitator during discussions. Participants were also able to distinguish participant profiles and names from those of the AI facilitator.

Research Context: We conducted our study in Afghanistan during a challenging period when the Taliban were in control. Given this context, we had to carefully design our research questions, measures, and procedures to ensure the safety of participants.

In our revised manuscript, we added a subsection on ethical considerations in methods, to provide a more comprehensive and transparent account of how we addressed these concerns. We believe that these measures align with ethical standards and best practices in conducting research involving human subjects, thereby contributing to the overall ethical integrity of our study.

R3 Comment 16.

- Was the study preregistered and if so, did the authors follow the preregistration? This is unclear from the manuscript.

Response

We did not preregister our study. We acknowledge the importance of preregistration in promoting transparency and rigor in research, and we will consider this aspect in our future research endeavors.

References

- 1 White, F. A., Turner, R. N., Verrelli, S., Harvey, L. J. & Hanna, J. R. Improving intergroup relations between Catholics and Protestants in Northern Ireland via E-contact. *European Journal of Social Psychology* **49**, 429-438 (2019).
- 2 Maunder, R. D., White, F. A. & Verrelli, S. Modern avenues for intergroup contact: Using E-contact and intergroup emotions to reduce stereotyping and social distancing against people with schizophrenia. *Group Processes & Intergroup Relations* **22**, 947-963 (2019).
- 3 Short, J., Williams, E. & Christie, B. *The social psychology of telecommunications*. (John Wiley & Sons, 1976).
- 4 Kim, S., Eun, J., Oh, C., Suh, B. & Lee, J. in *Proceedings of the 2020 CHI Conference on Human Factors in Computing Systems* 1–13 (Association for Computing Machinery, Honolulu, HI, USA, 2020).
- 5 Festinger, L. *A theory of cognitive dissonance*. (Stanford University Press, 1957).
- 6 Condra, L. N. & Linardi, S. Casual Contact and Ethnic Bias: Experimental Evidence from Afghanistan. *The Journal of Politics* **81**, 1028-1042 (2019).
<https://doi.org/10.1086/703380>
- 7 Zhou, Y.-Y. & Lyall, J. Prolonged Contact Does Not Reshape Locals' Attitudes toward Migrants in Wartime Settings: Experimental Evidence from Afghanistan. *Available at SSRN 3679746* (2022).
- 8 Pettigrew, T. F. & Tropp, L. R. A meta-analytic test of intergroup contact theory. *Journal of personality and social psychology* **90**, 751 (2006).
- 9 Imperato, C., Schneider, B. H., Caricati, L., Amichai-Hamburger, Y. & Mancini, T. Allport meets internet: A meta-analytical investigation of online intergroup contact and prejudice reduction. *International Journal of Intercultural Relations* **81**, 131-141 (2021).
- 10 Harwood, J. Indirect and mediated intergroup contact. *The International Encyclopedia of Intercultural Communication*, 1-9 (2017).
- 11 Schwab, A. K., Sagioglou, C. & Greitemeyer, T. Getting connected: Intergroup contact on Facebook. *The Journal of social psychology* **159**, 344-348 (2019).
- 12 Mancini, T. & Imperato, C. Can social networks make us more sensitive to social discrimination? E-contact, identity processes and perception of online sexual discrimination in a sample of Facebook users. *Social Sciences* **9**, 47 (2020).
- 13 White, F. A., Harvey, L. J. & Abu-Rayya, H. M. Improving intergroup relations in the Internet age: A critical review. *Review of General Psychology* **19**, 129-139 (2015).
- 14 White, F. A. & Abu-Rayya, H. M. A dual identity-electronic contact (DIEC) experiment promoting short-and long-term intergroup harmony. *Journal of Experimental Social Psychology* **48**, 597-608 (2012).
- 15 Amichai-Hamburger, Y. & McKenna, K. Y. The contact hypothesis reconsidered: Interacting via the Internet. *Journal of Computer-mediated communication* **11**, 825-843 (2006).
- 16 McKeown, S. & Dixon, J. The "contact hypothesis": Critical reflections and future directions. *Social and Personality Psychology Compass* **11**, e12295 (2017).
- 17 Dixon, J. C. The ties that bind and those that don't: Toward reconciling group threat and contact theories of prejudice. *Social forces* **84**, 2179-2204 (2006).
- 18 Taylor, M. C. How White Attitudes Vary with the Racial Composition of Local Populations: Numbers Count. *American Sociological Review* **63**, 512-535 (1998).
<https://doi.org/10.2307/2657265>
- 19 García-Faroldi, L. Determinants of Attitudes towards Immigration: Testing the Influence of Interculturalism, Group Threat Theory and National Contexts in Time of Crisis. *International Migration* **55**, 10-22 (2017).
<https://doi.org/https://doi.org/10.1111/imig.12261>

- 20 Blumer, H. Race Prejudice as a Sense of Group Position. *Pacific Sociological Review* **1**,
3-7 (1958). <https://doi.org/10.2307/1388607>
- 21 Douglas, K. M. & McGarty, C. Identifiability and self-presentation: Computer-mediated
communication and intergroup interaction. *British journal of social psychology* **40**, 399-
416 (2001).
- 22 Mousa, S. *Contact, conflict, and social cohesion*, Stanford University, (2020).
- 23 Asimovic, N., Nagler, J., Bonneau, R. & Tucker, J. A. Testing the effects of Facebook
usage in an ethnically polarized setting. *Proceedings of the National Academy of
Sciences* **118**, e2022819118 (2021).
- 24 Pearson, A. R. *et al.* The fragility of intergroup relations: Divergent effects of delayed
audiovisual feedback in intergroup and intragroup interaction. *Psychological Science* **19**,
1272-1279 (2008).
- 25 Kunz, W. & Rittel, H. W. Vol. Working Paper No. 131 (Institute of Urban and Regional
Development, University of California, Berkeley, California, 1970).
- 26 Ehsan, U., Liao, Q. V., Muller, M., Riedl, M. O. & Weisz, J. D. in *Proceedings of the 2021
CHI Conference on Human Factors in Computing Systems* Article 82 (Association for
Computing Machinery, Yokohama, Japan, 2021).
- 27 Lee, N., Madotto, A. & Fung, P. in *Wnlp@ Acl.* 177-180.
- 28 Bang, J., Kim, S., Nam, J. W. & Yang, D.-G. in *2021 International Conference on
Platform Technology and Service (PlatCon).* 1-5 (IEEE).
- 29 Mozafari, N., Hammerschmidt, M. & Weiger, W. in *Proceedings of the International
Conference on Information Systems.*
- 30 Adair, J. G. The Hawthorne effect: A reconsideration of the methodological artifact.
Journal of Applied Psychology **69**, 334-345 (1984). [https://doi.org/10.1037/0021-
9010.69.2.334](https://doi.org/10.1037/0021-9010.69.2.334)
- 31 Mousa, S. Building social cohesion between Christians and Muslims through soccer in
post-ISIS Iraq. *Science* **369**, 866-870 (2020).
<https://doi.org/doi:10.1126/science.abb3153>
- 32 Walther, J. B., Hoter, E., Ganayem, A. & Shonfeld, M. Computer-mediated
communication and the reduction of prejudice: A controlled longitudinal field experiment
among Jews and Arabs in Israel. *Computers in Human Behavior* **52**, 550-558 (2015).
- 33 Pertiwi, Y. G., Geers, A. L. & Lee, Y.-T. Rethinking intergroup contact across cultures:
Predicting outgroup evaluations using different types of contact, group status, and
perceived sociopolitical contexts. *Journal of Pacific Rim Psychology* **14**, e16 (2020).
- 34 Barlow, F. K. *et al.* The contact caveat: Negative contact predicts increased prejudice
more than positive contact predicts reduced prejudice. *Personality and social
Psychology bulletin* **38**, 1629-1643 (2012).
- 35 Cernat, V. Intergroup contact in Romania: When minority size is positively related to
intergroup conflict. *Journal of Community & Applied Social Psychology* **20**, 15-29 (2010).
<https://doi.org/https://doi.org/10.1002/casp.1001>
- 36 MacInnis, C. C. & Page-Gould, E. How Can Intergroup Interaction Be Bad If Intergroup
Contact Is Good? Exploring and Reconciling an Apparent Paradox in the Science of
Intergroup Relations. *Perspectives on Psychological Science* **10**, 307-327 (2015).
<https://doi.org/10.1177/1745691614568482>
- 37 Blalock, H. M. *Toward a Theory of Minority-Group Relations.* (Wiley, 1967).
- 38 De Coninck, D. Fear of Terrorism and Attitudes Toward Refugees: An Empirical Test of
Group Threat Theory. *Crime & Delinquency* **68**, 550-571 (2022).
<https://doi.org/10.1177/0011128720981898>
- 39 Jacobs, L., Boukes, M. & Vliegthart, R. Combined Forces: Thinking and/or Feeling?
How News Consumption Affects Anti-Muslim Attitudes through Perceptions and

- Emotions about the Economy. *Political Studies* **67**, 326-347 (2019).
<https://doi.org/10.1177/0032321718765696>
- 40 Millon, T. & Lerner, M. J. *Handbook of psychology volume 5 Personality and social*
psychology. (John Wiley & Sons, 2003).
- 41 Schlueter, E. & Scheepers, P. The relationship between outgroup size and anti-outgroup
attitudes: A theoretical synthesis and empirical test of group threat- and intergroup
contact theory. *Social Science Research* **39**, 285-295 (2010).
<https://doi.org/https://doi.org/10.1016/j.ssresearch.2009.07.006>
- 42 Dovidio, J. F., Gaertner, S. L. & Kawakami, K. Intergroup contact: The past, present, and
the future. *Group processes & intergroup relations* **6**, 5-21 (2003).
- 43 De Coninck, D., Rodríguez-de-Dios, I. & d'Haenens, L. The contact hypothesis during
the European refugee crisis: Relating quality and quantity of (in)direct intergroup contact
to attitudes towards refugees. *Group Processes & Intergroup Relations* **24**, 881-901
(2021). <https://doi.org/10.1177/1368430220929394>
- 44 Green, D. Immigrant Perception in Japan: A Multilevel Analysis of Public Opinion. *Asian*
Survey **57**, 368-394 (2017). <https://doi.org/10.1525/as.2017.57.2.368>
- 45 Green, D. & Kadoya, Y. Contact and Threat: Factors Affecting Views on Increasing
Immigration in Japan. *Politics & Policy* **43**, 59-93 (2015).
<https://doi.org/https://doi.org/10.1111/polp.12109>
- 46 Ellis, D. G. & Maoz, I. Online Argument between Israeli Jews and Palestinians. *Human*
Communication Research **33**, 291-309 (2007). <https://doi.org/10.1111/j.1468-2958.2007.00300.x>
- 47 Bogardus, E. S. Measuring social distance. *Journal of applied sociology* **9**, 299-308
(1925).
- 48 Stephan, W. G. & Stephan, C. W. Intergroup Anxiety. *Journal of Social Issues* **41**, 157-
175 (1985). <https://doi.org/https://doi.org/10.1111/j.1540-4560.1985.tb01134.x>
- 49 Swart, H., Hewstone, M., Christ, O. & Voci, A. Affective mediators of intergroup contact:
A three-wave longitudinal study in South Africa. *Journal of Personality and Social*
Psychology **101**, 1221-1238 (2011). <https://doi.org/10.1037/a0024450>

9th Nov 23

Dear Dr. Sahab,

Thank you for your patience during the peer-review process. Your manuscript titled "Reducing Interethnic Bias with Conversational Agent in E-contact Experiments in Afghanistan" has now been seen by 2 of the original reviewers, and I include their comments at the end of this message. They find your work of interest but raised some important points. We are interested in the possibility of publishing your study in Communications Psychology but would like to consider a final round of revisions for editorial evaluation, although we may send the work back for further peer review if necessary.

We therefore invite you to revise and resubmit your manuscript, along with a point-by-point response to the reviewers and our editorial requests. Please highlight all changes in the manuscript text file.

Editorially, we consider it important that the revised manuscript includes further specification of the specific statistical analysis targeted by the power analysis. Please indicate if the power analysis was conducted prior to data collection (i.e., a priori). If it did not target the effect size in the interaction contract, please provide us with a sensitivity analysis for this effect.

Please add an Ethics & Inclusion Statement to the manuscript clarifying if you sought an ethics agreement from an Afghan institution (and if not, why). You can find details on these policies here: <https://www.nature.com/nature-portfolio/editorial-policies/authorship#authorship-inclusion-and-ethics-in-global-research> Please comment on items 1-3, 5, 7-10. We appreciate that Afghanistan may be the country of origin for some of the co-authors, but it is currently unclear to us whether they resided in Afghanistan or were collaborating with an Afghan institution during the project.

In your data availability statement please provide more clarity on the ethical considerations and restrictions. Please add a code availability statement.

Please report Bayes Factors throughout the analyses. In interpreting BFs, those less than 3.0 but greater than 0.3 are considered inconclusive. You may state that there is no credible evidence for a difference but also that there is no decisive evidence for a lack of difference.

Please add confidence intervals as part of your stats reporting where they have not yet been reported.

In the discussion, claims regarding the influential role of the CA in enhancing participants' sense of social presence within the online discussion environment should be removed or presented as speculation. Your data only speak to differences in engagement and do not include evidence for "sense of social presence."

We encourage you to place the focus on contextualizing the current, not historical, situation in Afghanistan. Statements describing the current situation in Afghanistan, in particular about existing tensions between the groups (i.e., those immediately bearing on the research question), need to be supported by a) official demographic data or reports b) peer-reviewed research articles. As a general

guidance, please avoid relying on opinion pieces in the references to motivate specifics of the design of the study (incl. its setting).

Please note that your revised manuscript must comply with our formatting and reporting requirements, which are summarized in the following guide. I am attaching the formatting checklist to provide further guidance.

Communications Psychology formatting guide .

Please use the following link to submit your revised manuscript, point-by-point response to the referees' comments (which should be in a separate document to any cover letter) and the completed checklist:

[link redacted]

Please do not hesitate to contact me if you have any questions or would like to discuss these revisions further. We look forward to seeing the revised manuscript and thank you for the opportunity to review your work.

Best regards,

Jennifer Bellingtier

Jennifer Bellingtier, PhD
Senior Editor
Communications Psychology

EDITORIAL POLICIES AND FORMATTING

We ask that you ensure your manuscript complies with our editorial policies. Please ensure that the

following formatting requirements are met, and any checklist relevant to your research is completed and uploaded as a Related Manuscript file type with the revised article.

Editorial Policy: Policy requirements (Download the link to your computer as a PDF.)

* **CODE AVAILABILITY:** All Communications Psychology manuscripts must include a section titled "Code Availability" at the end of the methods section. In the event of publication, we require that the custom analysis code supporting your conclusions is made available in a publicly accessible repository; at publication, we ask you to choose a repository that provides a DOI for the code; the link to the repository and the DOI will need to be included in the Code Availability statement. Publication as Supplementary Information will not suffice. We ask you to prepare code at this stage, to avoid delays later on in the process.

* **DATA AVAILABILITY:**

All Communications Psychology manuscripts must include a section titled "Data Availability" at the end of the Methods section or main text (if no Methods). More information on this policy, is available at <http://www.nature.com/authors/policies/data/data-availability-statements-data-citations.pdf>.

At a minimum the Data availability statement must explain how the data can be obtained and whether there are any restrictions on data sharing. Communications Psychology strongly endorses open sharing of data. If you do make your data openly available, please include in the statement:

We recommend submitting the data to discipline-specific, community-recognized repositories, where possible and a list of recommended repositories is provided at <http://www.nature.com/sdata/policies/repositories>.

If a community resource is unavailable, data can be submitted to generalist repositories such as figshare or Dryad Digital Repository. Please provide a unique identifier for the data (for example a DOI or a permanent URL) in the data availability statement, if possible. If the repository does not provide identifiers, we encourage authors to supply the search terms that will return the data. For data that have been obtained from publicly available sources, please provide a URL and the specific data product name in the data availability statement. Data with a DOI should be further cited in the methods reference section.

REVIEWERS' EXPERTISE:

Reviewer #1 Intergroup contact

Reviewer #3 Intergroup contact

REVIEWERS' COMMENTS:

Reviewer #1 (Remarks to the Author):

The authors have conducted an excellent job with reviewing and responding to our comments. I truly appreciate the level of specificity and clarity that they put into this revision. As I said in my initial review, I really appreciate this study but my main concern still remains unanswered (despite excellent revisions). This issue refers to underlying mechanism of CA' facilitation effect. The authors do offer a possible explanation of CA mimicking the authority condition but it still remains speculative at this point. As I said, I think this is a solid paper as it is but I am just struggling with explaining the observed effects.

Reviewer #3 (Remarks to the Author):

The manuscript is a vast improvement on the original. Clearly, a lot of work went into the revision and it's paid off. My initial doubts and skepticism about this manuscript have been removed by the quality of the revision. The authors address my main concerns including focusing the hypotheses to be tested, ethical implications, and limitations of the design. I only have 2 minor concerns the authors may want to consider moving forward.

1. The power calculation resulting in a sample size of 128 seems improbably small/low. It seems that you would want to estimate sample size using repeated ANOVA to capture the time dimension of the 2x3 design.
2. The authors should note that the design was not pre-registered and explain why, given that pre-registration is increasingly the norm. In the end, the hypotheses have been significantly revised post data collection.

Congratulations again on a successful revision effort.

Reviewer 1

The authors have conducted an excellent job with reviewing and responding to our comments. I truly appreciate the level of specificity and clarity that they put into this revision. As I said in my initial review, I really appreciate this study but my main concern still remains unanswered (despite excellent revisions). This issue refers to underlying mechanism of CA' facilitation effect. The authors do offer a possible explanation of CA mimicking the authority condition but it still remains speculative at this point. As I said, I think this is a solid paper as it is but I am just struggling with explaining the observed effects.

Response

We deeply appreciate your positive feedback on our revisions. Your insightful comments have been invaluable throughout this review process. We also want to express our gratitude for your continued engagement with the manuscript and your thoughtful review.

We fully understand and share your concern regarding the explanation of the underlying mechanism of the Conversational Agent (CA)'s facilitation effect. Despite our efforts to offer a plausible explanation, we acknowledge that the current interpretation remains speculative.

Our study's initial hypothesis posited that CA facilitation would reduce intergroup anxiety, thereby mediating intergroup prejudice (Pettigrew and Tropp 2006, White and Abu-Rayya 2012, Abu-Rayya 2017). While we did observe a significant reduction in intergroup anxiety at time 2 and time 3 in the treatment group, our mediation analysis did not yield significant results.

Acknowledging the complexities involved in mediation analysis, particularly in longitudinal studies, we are committed to enhancing our understanding of the CA's facilitation mechanism. In future studies, we plan to incorporate additional subjective and objective measures, exploring potential mediators and moderators, as well as conducting a more comprehensive analysis of social presence and its impact on intergroup dynamics.

Your guidance is instrumental in shaping the direction of our research, and we are committed to further refining our methodologies to offer a more nuanced and substantiated account of the observed effects.

Thank you once again for your insightful comments and continued support.

Reviewer 3

The manuscript is a vast improvement on the original. Clearly, a lot of work went into the revision and it's paid off. My initial doubts and skepticism about this manuscript have been removed by the quality of the revision. The authors address my main concerns including focusing the hypotheses to be tested, ethical implications, and limitations of the design. I only have 2 minor concerns the authors may want to consider moving forward.

Response.

We appreciate your thorough evaluation of the revised manuscript and are encouraged by your positive assessment. Your constructive feedback has played a crucial role in shaping the improvements made during the revision process. We are grateful for the time and effort you have dedicated to the manuscript, and your insights have been instrumental in refining the study.

We are committed to addressing the remaining concerns you have raised, along with those from other reviewers and editors, to ensure that the manuscript meets the standards of clarity and validity. Your feedback serves as a valuable guide, and we look forward to further enhancing the manuscript based on the collective suggestions.

R3 Comment 1.

The power calculation resulting in a sample size of 128 seems improbably small/low. It seems that you would want to estimate sample size using repeated ANOVA to capture the time dimension of the 2x3 design.

Response

Thank you for addressing the concern about the sample size determination. We appreciate the opportunity to clarify this aspect of our study.

Indeed, we conducted power analyses, accounting for the time dimension using repeated measures ANOVA, and we have now included this detail in the manuscript (lines 352-353). We recognize that introducing the time dimension resulted in a smaller suggested sample size compared to what might be recommended with traditional independent sample t-tests without considering the longitudinal aspect.

Given our unique study context, along with budget and time constraints, aligning our sample size with our longitudinal research design was feasible within the broader considerations of our study's specific circumstances.

In future research endeavors, we plan to carefully consider these factors and explicitly weigh the trade-offs between sample size and research design.

R3 Comment 2.

The authors should note that the design was not pre-registered and explain why, given that pre-registration is increasingly the norm. In the end, the hypotheses have been significantly revised post data collection.

Congratulations again on a successful revision effort.

Response

We appreciate the opportunity to address the issue of pre-registration and the significant revisions made to the hypotheses post data collection. Given the growing emphasis on pre-registering research designs for transparency and rigor, we acknowledge the value of this practice. However, our decision not to pre-register was influenced by specific challenges in our study in Afghanistan, particularly the sensitive nature of ethnic issues and the unpredictable research environment under the Taliban government, which introduced uncertainties about our ability to conduct the research.

Despite these complexities, we want to underscore our commitment to transparency and reproducibility in research and assure you that we will earnestly consider pre-registration in future research endeavors. In response to your valuable feedback, we have provided clarification about the decision not to pre-register and its reasons, along with details on revising the hypotheses in the “Ethical Considerations” section of our “Methods” (lines 573-583). Your insights played a crucial role in guiding these revisions, and we believe they enhance the accuracy and clarity of our reported results.

We remain steadfast in upholding the highest standards of research integrity and appreciate the opportunity to bolster the robustness of our study through the peer review process. Thank you for your time and thoughtful consideration.

References

Abu-Rayya, H. M. (2017). "Majority members' endorsement of the acculturation integrationist orientation improves their outgroup attitudes toward ethnic minority members: An electronic-contact experiment." Computers in Human Behavior **75**: 660-666.

Pettigrew, T. F. and L. R. Tropp (2006). "A meta-analytic test of intergroup contact theory." Journal of personality and social psychology **90**(5): 751.

White, F. A. and H. M. Abu-Rayya (2012). "A dual identity-electronic contact (DIEC) experiment promoting short-and long-term intergroup harmony." Journal of Experimental Social Psychology **48**(3): 597-608.

24th Jan 24

Dear Dr Sahab,

Your manuscript titled "Reducing Interethnic Bias with Conversational Agent in E-contact Experiments in Afghanistan" has now been seen by our reviewers, whose comments appear below. In light of their advice I am delighted to say that we are happy, in principle, to publish a suitably revised version in Communications Psychology under the open access CC BY license (Creative Commons Attribution v4.0 International License).

We therefore invite you to revise your paper one last time to address the remaining concerns of our reviewers and a list of editorial requests. At the same time we ask that you edit your manuscript to comply with our format requirements and to maximise the accessibility and therefore the impact of your work.

EDITORIAL REQUESTS:

Please review our specific editorial comments and requests regarding your manuscript in the attached "Editorial Requests Table" and attached Article file. Please outline your response to each request in the right hand column. Please upload the completed table with your manuscript files as a Related Manuscript file.

We feel that as a result of the comprehensive revisions and your efforts to address all referee requests, the introduction does not give the best possible succinct overview of the work that motivates the research question. We therefore suggest including some parts of the material that was added in response to the constructive referee comments in the Discussion and to cut redundancies.

SUBMISSION INFORMATION:

OPEN ACCESS:

Communications Psychology is a fully open access journal. Articles are made freely accessible on publication under a CC BY license (Creative Commons Attribution 4.0 International License). This license allows maximum dissemination and re-use of open access materials and is preferred by many research funding bodies.

For further information about article processing charges, open access funding, and advice and support from Nature Research, please visit <https://www.nature.com/commspsychol/article-processing-charges>

At acceptance, you will be provided with instructions for completing this CC BY license on behalf of all authors. This grants us the necessary permissions to publish your paper. Additionally, you will be asked to declare that all required third party permissions have been obtained, and to provide billing information in order to pay the article-processing charge (APC).

* TRANSPARENT PEER REVIEW: Communications Psychology uses a transparent peer review system. On author request, confidential information and data can be removed from the published reviewer reports and rebuttal letters prior to publication. If you are concerned about the release of confidential data, please let us know specifically what information you would like to have removed. Please note that we cannot incorporate redactions for any other reasons.

* CODE AVAILABILITY: All Communications Psychology manuscripts must include a section titled "Code Availability" at the end of the methods section. We require that the custom analysis code supporting your conclusions is made available in a publicly accessible repository at this stage; please choose a repository that generates a digital object identifier (DOI) for the code; the link to the repository and the DOI must be included in the Code Availability statement. Publication as Supplementary Information will not suffice.

* DATA AVAILABILITY:

[link redacted]

Best regards,

Jennifer Bellingtier

Jennifer Bellingtier, PhD
Senior Editor
Communications Psychology